# DOMAIN GENERALIZATION IN-THE-WILD: DISENTANGLING CLASSIFICATION FROM DOMAIN-AWARE REPRESENTATIONS

## ABSTRACT

Evaluating domain generalization (DG) for foundational models like CLIP is challenging, as web-scale pretraining data potentially covers many existing benchmarks. Consequently, current DG evaluation may neither be sufficiently challenging nor adequately test genuinely unseen data scenarios. To better assess the performance of CLIP on DG in-the-wild, a scenario where CLIP encounters challenging unseen data, we consider two approaches: (1) evaluating on 33 diverse datasets with quantified out-of-distribution (OOD) scores after fine-tuning CLIP on ImageNet, and (2) using unlearning to make CLIP 'forget' some domains as an approximation. We observe that CLIP's performance deteriorates significantly on more OOD datasets. To address this, we present CLIP-DCA (**D**isentangling **C**lassification from enhanced domain **A**ware representations). Our approach is motivated by the observation that while standard domain invariance losses aim to make representations domain-invariant, this can be harmful to foundation models by forcing the discarding of domain-aware representations beneficial for generalization. We instead hypothesize that enhancing domain awareness is a prerequisite for effective domain-invariant classification in foundation models. CLIP-DCA identifies and enhances domain awareness within CLIP's encoders using a separate domain head and synthetically generated diverse domain data. Simultaneously, it encourages domain-invariant classification through disentanglement from the domain features. CLIP-DCA shows significant improvements within this challenging evaluation compared to existing methods, particularly on datasets that are more OOD.

## 1 INTRODUCTION

Domain generalization (DG) aims to train models that maintain robust performance when encountering out-of-distribution (OOD) data (Zhou et al., 2022a). A key assumption of DG is that the target domains represent novel data distributions for evaluation. However, this assumption is challenged when evaluating pretrained foundation models like CLIP (Radford et al., 2021) and ALIGN (Jia et al., 2021). These models have been trained on comprehensive web-scale datasets, thus have likely been exposed to most existing domains, contributing to its impressive zero-shot capabilities. Consequently, much research has focused on adapting CLIP through parameter-efficient fine-tuning (Zhou et al., 2022c;b; Gao et al., 2024; Zhang et al., 2022), regularization using the original weights (Wortsman et al., 2022; Nam et al., 2024; Oh et al., 2024; Shu et al., 2023), and even transductive methods (Wallingford et al., 2023; Martin et al., 2024), largely preserving its pretrained knowledge. However, this reliance on pretrained knowledge is predicated on an assumption of true OOD robustness that is now being challenged. Recent studies show that evaluating foundation models for DG is often compromised by data leakage

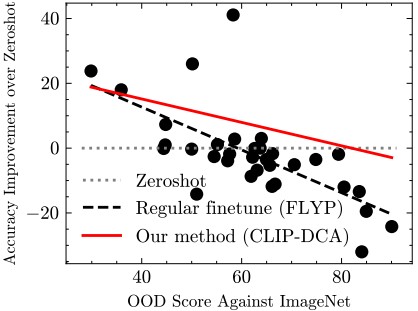

Figure 1: Improvement over zeroshot after finetuning on ImageNet (in %). Each dot represents a target dataset. OOD scores are quantified relative to ImageNet (source dataset), illustrating the challenge of DG in-the-wild.

from web-scale pre-training (Teterwak et al., 2024; Yu et al., 2024). Teterwak et al. (2024) addresses this by analyzing generalization from learned to unlearned samples within the pre-training data, while Yu et al. (2024) proposes training models from scratch to avoid contamination entirely. These studies, along with findings that retraining CLIP on cleaner data degrades OOD performance, suggest that current DG evaluations may overestimate true OOD robustness (Mayilvahanan et al., 2024). While these studies provide critical insight, they do not offer a tractable setting to test the generalization performance of CLIP after specific, contaminating knowledge has been selectively removed.

To address this gap, we propose that DG evaluation for foundation models, such as CLIP, should be more challenging, to approximate "domain generalization in-the-wild," where CLIP might encounter diverse and challenging new data in the real-world. We evaluate CLIP on 33 target datasets spanning a diverse range of OODness. To systematically approach evaluation, we quantify a multi-modal OOD score (Sec. 3.2), using ImageNet as both an anchor and a source dataset owing to its inclusion of many classes and concepts. We find that after finetuning on ImageNet, CLIP's DG performance degrades on datasets with higher OOD scores with respect to ImageNet (Figure 1), consistent with the domain contamination findings (Mayilvahanan et al., 2024). In addition, to further simulate truly unseen domains, we use an unlearning technique (Sepahvand et al., 2025) to make CLIP forget some domains (Sec. 3.3), and find significant performance degradation for existing robust finetuning methods.

Our results (Figure 9), alongside findings on domain contamination (Mayilvahanan et al., 2024), suggest that for DG in-the-wild, different robust finetuning algorithms are needed for genuinely unseen data. In light of this, we present CLIP-DCA (**D**isentangling **C**lassification from enhanced domain **A**ware representations), an end-to-end finetuning method to improve the robustness of CLIP on truly OOD data. A key idea in DG is that learning domain-invariant features is beneficial for robust generalization (Zhou et al., 2022a; Ganin et al., 2016). However, naively enforcing domain invariance for a pretrained foundation model could cause catastrophic forgetting of useful features learned from diverse domains during pretraining as the model is forced to make its representations entirely domain-invariant. We hypothesize that to learn effective domain invariance, domain awareness is a prerequisite. This awareness is critical to maintain CLIP's vast knowledge, which includes generalizable features that support capabilities like zero-shot classification. By enhancing domain awareness, CLIP can also selectively disentangle classification from domain-specific aspects, thereby achieving robust generalization without forgetting valuable information.

We combine the idea of domain awareness and domain invariance by encouraging them simultaneously within CLIP-DCA (Figure 2). Specifically, we encourage domain awareness within CLIP's image and text encoders, while promoting domain invariance specifically at the final classification layer through disentanglement. Our premise is that while domain awareness is a requirement to maintain pre-existing knowledge, this awareness can be disentangled for domain-invariant classification and robust generalization. To achieve this, we add a new head to the CLIP image encoder, called the domain head, which is trained to understand domains. The original classification head is then disentangled from the domain head, effectively learning domain awareness within its encoders and achieving domain invariance at the classification stage. Additionally, since many datasets lack distinct domains or textual descriptions, and the definition of 'domain' is often vague in DG in-the-wild, we address this by using diffusion models to create images of artificial domains and Multimodal LLMs (MLLMs) to generate descriptions for these artificial domains (Sec. 2.2). Our contributions are summarized as follows:

- We demonstrate potential limitations in current DG evaluations of foundation models, supported by our results and recent studies. Existing benchmarks may overestimate true OOD robustness, potentially leading finetuning strategies towards in-distribution improvement rather than OOD.
- We propose more challenging and holistic evaluations for DG in-the-wild. We use an expanded cross-dataset evaluation setting spanning 33 datasets from diverse domains, indexed by multi-modal OOD scores. We also use an unlearned model to further approximate unseen domains.
- We introduce CLIP-DCA, a novel finetuning method that improves OOD robustness by disentangling classification from enhanced domain-aware representations. We find that on more OOD target datasets, CLIP-DCA performs significantly better compared to existing robust finetuning methods, while performance is similar across all methods on less OOD target datasets.

**Related Work.** A comprehensive review is in Appendix A. Domain generalization (DG) has traditionally focused on learning domain-invariant representations (Ganin et al., 2016; Zhou et al.,

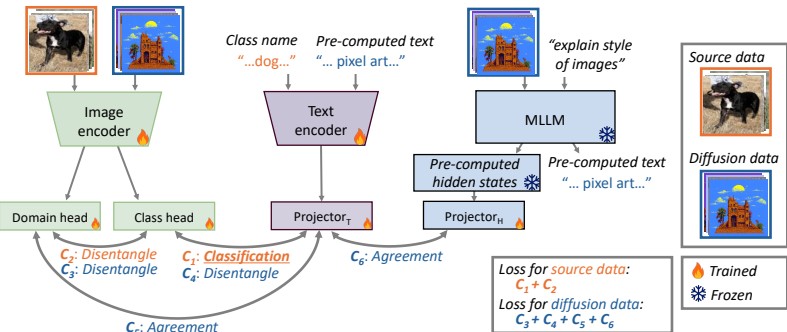

Figure 2: CLIP-DCA applies different sets of losses to source data images and diffusion images. For source images, accurate classification is encouraged through the classification loss between class head and text encoder ($C_1$). Invariance is encouraged through the disentanglement between domain and class heads ($C_2$). With diffusion images, domain invariance is encouraged through the disentanglement between the domain and class heads ($C_3$), and disentanglement between class head and text encoder ($C_4$). Domain awareness is encouraged through the agreement between the domain head and the text encoder ($C_5$), and the agreement between the text encoder and the MLLM hidden states ($C_6$). During inference, only the class head and text projector are used for classification.

2022a), but naively applying these methods to foundation models like CLIP can cause catastrophic forgetting. Consequently, most robust CLIP finetuning methods aim to preserve pretrained knowledge through parameter-efficient finetuning (PEFT) (Zhou et al., 2022c; Gao et al., 2024) or regularization towards the original weights (Wortsman et al., 2022; Shu et al., 2023). However, this reliance on pretrained knowledge is being questioned by recent findings of domain contamination in web-scale datasets (Teterwak et al., 2024; Yu et al., 2024; Mayilvahanan et al., 2024), which suggest current evaluations may overestimate true OOD robustness. Our work addresses this by proposing a more challenging evaluation framework and a method that learns targeted invariance without sacrificing pretrained knowledge.

## 2 CLIP-DCA: DISENTANGLING CLASSIFICATION FROM ENHANCED DOMAIN-AWARE REPRESENTATIONS

To address the challenges of DG in-the-wild, we introduce CLIP-DCA (**D**isentangling **C**lassification from enhanced domain **A**ware representations), a finetuning method designed to improve robustness on genuinely unseen data.

### 2.1 ENCOURAGING DOMAIN AWARENESS AND INVARIANCE SIMULTANEOUSLY

Our key hypothesis is that domain invariance at the decision-making stage is beneficial for generalizing to unseen domains. At the same time, domain awareness is required for retaining the vast pretrained knowledge of CLIP. We achieve them simultaneously by encouraging domain awareness in the encoders, while enforcing domain invariance only in the classifier of CLIP through disentanglement. The intuition is that **if a model understands what constitutes as domain-specific features, then it can learn to disregard it appropriately during classification on unseen domains**.

Enforcing domain invariance in the encoder through conventional domain adversarial learning, for instance, can be harmful. Our experiments show that applying invariance directly leads to worse performance compared to standard finetuning (Figure 8). Forcing the entire model to become domain-invariant can lead to the forgetting of valuable, fine-grained features learned during the pretraining on a large dataset. Conversely, existing CLIP robust finetuning methods discourage divergence from the original pretrained model, and rely on the assumption that CLIP is inherently robust to OOD data. This assumption is challenged by our results (Figure 9) and evidence for domain contamination (Teterwak et al., 2024; Yu et al., 2024; Mayilvahanan et al., 2024).

Instead, we focus on enforcing domain invariance only at the final classification layer, while simultaneously encouraging the image encoder to become domain-aware. Our intuition is that a comprehensive understanding of various domains enables the model to more effectively disregard domain-specific

influences during inference. The diverse set of generated diffusion images and their descriptions (detailed in Section 2.2) provides the necessary signals for enhancing this domain awareness.

To implement this, we introduce an architectural addition to the CLIP image encoder. We add an additional linear projection head, termed the image domain head ($I_D$), which has the same dimensionality as the original image projection head, referred to as the image class head ($I_C$), as shown in Figure 2. We do not add a corresponding domain head to the text encoder for two reasons. First, in most downstream classification datasets, only class names are available as text inputs, without domain descriptions. Second, textual information inherently allows for easier separation of domain and class attributes. For instance, a prompt like "a sketch of a dog" clearly distinguishes class ("dog") from domain ("sketch"). Note that for inference, the standard pipeline is used as shown in Figure 3. The domain head and other losses are not used.

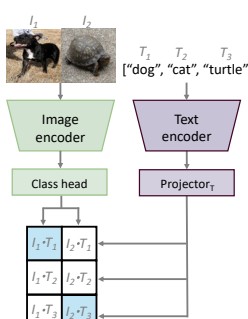

Figure 3: Standard CLIP inference pipeline using a dot product between image and text embeddings for classification.

During training, we use two distinct loss functions for the two types of data we use - the source dataset and generated diffusion images. We use $\ell_a$ to refer to agreement loss (the standard CLIP contrastive loss (Radford et al., 2021) or finetuning (Goyal et al., 2023)). We use $\ell_d$ to refer to disentanglement, which enforces statistical independence between two sets of representations. Inspired by the simplicity of self-supervised methods (Zbontar et al., 2021; Bardes et al., 2021), we achieve this by minimizing the correlation between the class and domain embeddings. The loss is formulated as the squared sum of the diagonal of the cross-correlation matrix between the batch-normalized class embeddings and domain embeddings. This penalizes any shared information, encouraging the class head to find predictive representations that are independent of features useful for domain prediction.

The role of the disentanglement loss is to enforce this separation. The underlying assumption is that if the two representations are truly disentangled, the features from the class head for a given sample should be statistically independent from the features learned by the domain head for that same sample. By minimizing the correlation between the class and domain embeddings, this loss encourages the class head to find representations that are predictive of the class label without using features that are also useful for predicting the domain. Conversely, it encourages the domain head to focus only on domain-specific information, as any shared information with the class head is penalized.

We simultaneously encourage accurate classification, domain awareness in both text and image encoders, and domain invariance at the classification stage with the following loss terms:

1. **For the source dataset images (e.g., ImageNet, with only class labels):**
   - A *classification loss* (i.e., the standard CLIP contrastive loss (Goyal et al., 2023)) between the output of the image class head and the text embedding of the class name, $C_1 := \ell_a(I_C, P_T)$.
   - A *disentanglement loss* between the class and domain heads, $C_2 := \ell_d(I_C, I_D)$.
   - For source dataset images, we minimize the loss function $\mathcal{L}_{source} = C_1 + C_2$.
2. **For the diffusion images and their MLLM-generated style descriptions:**
   - A *disentanglement loss* between the class head and domain head, $C_3 := \ell_d(I_C, I_D)$.
   - A *disentanglement loss* between the text embedding of style descriptions and the image class head to further encourage the class head to learn domain invariance, $C_4 := \ell_d(P_T, I_C)$.
   - An *agreement loss* between the output of the image domain head and the text embedding of the style description, enhancing domain head's domain awareness, $C_5 := \ell_a(P_T, I_D)$.
   - An *agreement loss* between the text embedding and the corresponding projected MLLM hidden state, enhancing the text encoder's domain awareness, $C_6 := \ell_a(P_T, P_H)$.
   - For diffusion images, we minimize the loss function $\mathcal{L}_{diffusion} = C_3 + C_4 + C_5 + C_6$.

For a detailed implementation, pseudocode for the main training loop is provided in Appendix B.

## 2.2 GENERATING DIVERSE DOMAINS

Traditional DG benchmarks provide multi-domain datasets, enabling the learning of domain invariance. However, our evaluation setup, which involves finetuning on a single source dataset like

ImageNet, lacks explicit multiple source domains, especially as the boundary for different domains becomes more vague for DG in-the-wild. Additionally, we hypothesize that to understand what constitutes as domain-specific features, a diverse number of domains are required.

To address this, we construct a small dataset with a diverse number of domains. As illustrated in Figure 4, we prompt a MLLM, specifically LLaVA (Liu et al., 2023), to generate ideas of 512 distinct styles for images (e.g. "pixel art"). The complete list of styles is available in Table G.1 in the Appendix. A text-to-image diffusion model (Stable Diffusion 3 (Esser et al., 2024)) then generates images from these stylistic prompts. We intentionally omit any class labels during image generation to ensure the styles are not biased towards specific classes. We generate 8 images per style, creating a dataset of 4096 images. Finally, the same MLLM generates textual domain descriptions (captions) for each style. We also store the hidden state representations from the MLLM that were used to generate these style descriptions, as these will be used to encourage domain awareness in the text encoder. The exact prompts used for style and description generation are detailed in Appendix F.

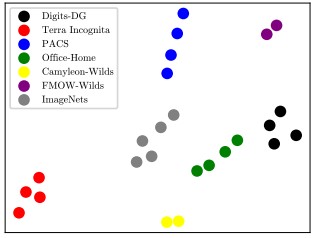

Figure 4: Pipeline for generating synthetic domain images and descriptions.

## 3 EXPERIMENTAL SETUP

### 3.1 EVALUATING DG IN-THE-WILD PERFORMANCE

We first analyze standard Domain Generalization (DG) benchmarks and find their domains are not well-separated. Using a Spectral-normalized Neural Gaussian Process (SNGP) (Liu et al., 2020) to compute pairwise OOD scores, we observe strong intra-benchmark clustering, as visualized in Figure 5. This clustering, along with CLIP's high zero-shot accuracy and the success of transductive methods on these datasets (Wallingford et al., 2023; Martin et al., 2024), suggests that current DG evaluations are not sufficiently challenging for large-scale models, possibly due to pre-training data contamination.

Figure 5: PCA visualization of domains from different domain generalization datasets

To address this, we finetune CLIP on ImageNet-1K (Deng et al., 2009) and evaluate its generalization capabilities across a more diverse benchmark of 33 target datasets spanning standard DG benchmarks and other challenging classification tasks (full list provided in Table E.1). A cross-dataset evaluation is significantly more challenging compared to traditional DG setups, as it involves larger visual distribution shifts and also shifts in class labels. This evaluation also aligns with the methodologies of prior studies investigating robust CLIP finetuning (Zhou et al., 2022c;b; Gao et al., 2024; Shu et al., 2023), while adding a broader coverage of domains. We use the CLIP ViT-B/32 model for all experiments. Further implementation details, including optimizer settings and specific hyperparameters for our method, are provided in Appendix C and D.

### 3.2 MEASURING OODNESS OF THE TARGET DATASETS

Given that our DG in-the-wild evaluation includes many target datasets with varying degrees of OODness compared to ImageNet, establishing a quantitative OOD metric is beneficial for a more holistic assessment of OOD robustness. A unique consideration for CLIP is its dual-encoder architecture. To provide a comprehensive score, we utilize OOD measures for both the image and text modalities. For the image encoder, we use SNGP (Liu et al., 2020) calibrated on the ImageNet validation data to compute an OOD score for all 33 target datasets. In addition, we use a text-based OOD measure (Fort et al., 2021) to measure OODness of class labels. This involves calculating classification probabilities on a combined label set of target

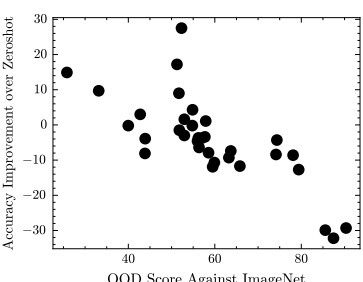

Figure 6: OOD score of 33 target datasets against ImageNet and classification accuracy improvement over zeroshot

dataset class names and ImageNet class names using the target domain image embeddings. The text OOD score is the summed probability assigned to the target-specific class names.

We verify that our OOD score shows a strong negative correlation (r=-0.756, p<0.001) with performance on target datasets after finetuning, as shown in Figure 6. Notably, we find that averaging the image and text OOD scores is important for accurately predicting post-finetuning accuracy. Relying solely on the image OOD score (r=-0.099) or the text OOD score (r=-0.608) yields weaker correlations, providing evidence that OOD scores in both modalities are necessary for a comprehensive understanding of OOD challenges in the context of CLIP. A detailed breakdown of the OOD scores for all 33 target datasets is provided in Appendix J.

## 3.3 SIMULATING UNSEEN DOMAINS VIA UNLEARNING

Retraining a foundation model like CLIP from scratch to omitting specific domains is computationally prohibitive. To overcome this, we use an unlearning method as a proxy to approximate a model that is not contaminated with domains relevant to our evaluation. This controlled experiment allows us to answer a critical research question: *How do robust fine-tuning methods perform on genuinely unseen domains?* Our results expose weaknesses in existing approaches that rely on pretrained weights.

Specifically, we adapt the adversarial learning-based unlearning method (Sepahvand et al., 2025) for domain forgetting. We finetune CLIP (Goyal et al., 2023) using a dual objective. First, to retain general knowledge, we train on a 595,000-image subset of the CC3M dataset (Sharma et al., 2018), referred to as GCC, previously used in LLaVA pretraining (Liu et al., 2023), serving as a manageable proxy for CLIP's original training data. Second, to approximate a scenario where domains similar to DomainNet are removed, we apply domain adversarial training (Ganin et al., 2016) on the DomainNet dataset, which we exclude from our target datasets. We attach a binary classifier to the penultimate layer of the image encoder. During training batches, this classifier is fed representations of random noise (assigned label 0) and images from DomainNet (assigned label 1). The gradient reversal layer (Ganin et al.,

Table 1: Unlearning effectiveness. ZS: Original zero-shot performance. FT: Baseline fine-tuning on the GCC retention set. Unlearn: Full unlearning combining retention on GCC with adversarial unlearning on DomainNet.

| Metric/Data | ZS | FT | Unlearn |
|---|---|---|---|
| ***Imagenet*** | | | |
| IN 1 | 54.2 | 52.0 | 48.8 |
| IN 2 | 48.4 | 45.5 | 41.8 |
| IN Sketch | 32.3 | 31.5 | 30.7 |
| IN A | 26.2 | 19.0 | 18.2 |
| IN R | 59.7 | 56.8 | 52.7 |
| ***DomainNet*** | | | |
| Clipart | 64.3 | 67.0 | 53.0 |
| Infograph | 41.6 | 41.0 | 34.0 |
| Painting | 54.4 | 53.9 | 47.0 |
| Real | 80.5 | 80.7 | 73.3 |
| Sketch | 57.9 | 57.2 | 45.5 |
| Quickdraw | 12.1 | 8.2 | 0.3 |
| Avg. on 33 | 51.1 | 49.7 | 45.5 |

2016) forces the image encoder to learn representations that confuse this classifier, making embeddings of DomainNet images and random noise indistinguishable, thereby encouraging the model to unlearn domain-specific features from DomainNet. The unlearning occurs concurrently with standard training on the GCC dataset to preserve CLIP's core capabilities. A pseudocode of the unlearning process is in Appendix B.

We deliberately unlearn on DomainNet, a dataset we do not use for final evaluation. Unlearning our target evaluation datasets directly would unfairly penalize baseline methods. Many methods are designed to regularize against large deviations from the original pretrained weights. By using DomainNet as a proxy for domain contamination, we ensure a fairer comparison. The effectiveness of our unlearning is confirmed by a performance drop on DomainNet while performance on many other datasets is largely retained (Table 1).

This experimental setup is distinct from other recent proposals. While Teterwak et al. (2024) separate samples based on whether they were learned during pre-training, their focus is on generalization from well-learned to seen-but-unlearned concepts. In contrast, Yu et al. (2024) evaluate domain generalization by training models from scratch without web-scale pre-training. Our approach is a unique and practical middle ground. We measure the performance of a model that benefits from web-scale pre-training but has had specific domain knowledge removed. This allows us to more directly isolate the effect of domain contamination on robust fine-tuning.

# 4 RESULTS AND DISCUSSION

## 4.1 FINETUNING ORIGINAL PRETRAINED CLIP

We first evaluate CLIP-DCA in the context of our domain generalization in-the-wild setup, using the original pretrained CLIP weights as the starting point. As shown in Figure 7, CLIP-DCA consistently improves performance over standard finetuning across target datasets. Importantly, the best-fit line for CLIP-DCA shows a flatter slope, indicating that it is more robust to more severe OOD data compared to regular finetuning. This observation aligns with our hypothesis that encouraging domain invariance at the decision-making layer, while simultaneously encouraging domain awareness within the encoders, is crucial for robust classification on unseen distributions.

Figure 8 provides a broader comparison against additional baselines. We observe that conventional domain adversarial learning (DANN (Ganin et al., 2016)), is harmful for CLIP, showing inferior performance compared to regular finetuning. This shows the potential disadvantage of enforcing domain invariance across the entire image encoder, which can lead to excessive forgetting of features learned during pretraining. This suggests the importance of approaches such as our proposed learning of targeted inv

Interestingly, on the most extremely OOD datasets, parameter-efficient finetuning (PEFT) techniques like CoOp (Zhou et al., 2022c) and CLIP-Adapter (Gao et al., 2024) perform best. PEFT methods minimally change a small subset of the original CLIP weights. Consequently, their performance shows much lower variance across the datasets, with improvements (around 1-2%). It is important to note that on extreme OOD datasets, all end-to-end finetuning methods exhibit lower performance than the zero-shot CLIP baseline. While CLIP-DCA mitigates this performance drop compared to standard finetuning, it does not entirely overcome it.

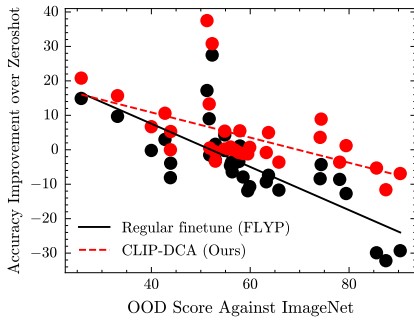

Figure 7: Performance comparison of CLIP-DCA against regular finetuning. Best-fit lines, determined by linear regression, illustrate performance trends.

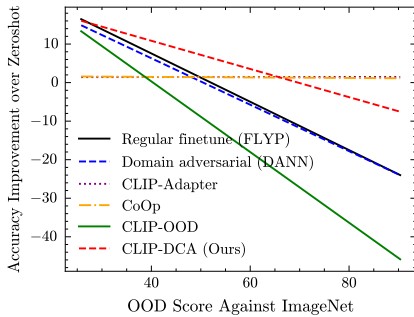

Figure 8: Comparison against more baselines.

This strong zero-shot performance has often been attributed to CLIP's inherent OOD generalization capability. However, the study by (Mayilvahanan et al., 2024) challenges this assumption and shows that this generalization could be attributed to domain contamination. They show that when CLIP is retrained solely on natural images, its OOD performance drops to similar levels as models trained exclusively on ImageNet. This drop could offer a plausible explanation for observations like those motivating Wise-FT (Wortsman et al., 2022), where standard finetuning was found to degrade OOD performance.

## 4.2 FINETUNING AFTER UNLEARNING

To further investigate the impact of potential domain contamination and to establish a more rigorously "unseen" evaluation, we applied the unlearning procedure detailed in Section 3.3 to the pretrained CLIP model. We then finetuned this "unlearned" model on ImageNet-1K and evaluated its performance. Table 2 shows the accuracies on the ImageNet variant datasets. Full per-dataset accuracy details for all methods, both before and after unlearning, are provided in Appendix I. For this analysis, we also include several end-to-end robust finetuning methods that add a linear classifier to CLIP. Due to their architecture, these specific baselines are evaluated only on the ImageNet variants as they cannot be adapted to datasets with different class labels.

Our results show that robust end-to-end finetuning methods remain effective for datasets that are less OOD even after unlearning. For instance, MIRO (Cha et al., 2022) and Wise-FT (Wortsman et al., 2022) outperform regular finetuning on ImageNet-V1, ImageNet-V2, and ImageNet-Sketch. To

Table 2: Accuracy on ImageNet variants

| Method | V1 | V2 | Sketch | A | R |
|--------|----|----|--------|---|---|
| Zeroshot (unlearned) | 48.8 | 41.8 | 30.7 | 18.2 | 52.7 |
| Regular Finetune | 69.8 | 58.4 | 34.7 | 15.0 | 52.6 |
| DANN | 70.0 | 58.2 | 33.2 | 16.5 | 52.0 |
| CLIP Adapter | 52.9 | 45.7 | 28.4 | 15.0 | 51.6 |
| CoOp | 53.3 | 46.2 | 29.1 | 16.1 | 52.8 |
| MMA | 71.7 | 60.0 | 36.1 | 7.8 | 37.0 |
| LwEIB | 53.8 | 46.7 | 30.4 | 16.3 | 54.1 |
| Wise-FT | 72.9 | 61.3 | 40.0 | 9.4 | 43.0 |
| MIRO | 74.1 | 62.7 | 35.7 | 7.3 | 33.2 |
| CLIP-OOD | 69.0 | 58.2 | 35.3 | 15.0 | 45.8 |
| **CLIP-DCA (Ours)** | **75.1** | **63.9** | **42.2** | **22.9** | **62.2** |

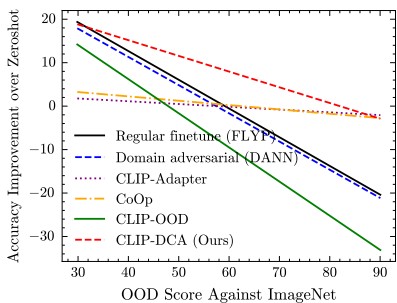

Figure 9: Comparison against baselines after unlearning.

broaden our comparison, we also include other recent CLIP-based DG methods such as MMA (Yang et al., 2024) and LwEIB (Yang et al., 2025), which similarly demonstrate that performance does not consistently generalize to more OOD datasets. However, consistent with the trends seen with the non-unlearned model, performance significantly drops on datasets with larger OOD scores, such as ImageNet-A and ImageNet-R. Similarly, PEFT methods show slight improvements over zero-shot on ImageNet-V1, V2, and Sketch, but their performance drops on ImageNet-A and R.

Figure 9 shows that the performance of all methods, even PEFT methods, further drops as OODness increases across target datasets when finetuning the unlearned model. If the unlearning process successfully reduced the knowledge of target-like domains, existing robust finetuning methods, which rely on the pretrained weights, would struggle on genuinely OOD data. These results suggest that our unlearning approach was effective in simulating a less contaminated starting point.

With the unlearned model, CLIP-DCA shows high performance. For datasets with moderate OOD scores relative to ImageNet, CLIP-DCA achieves larger performance improvements compared to other methods. More importantly, on the extremely OOD datasets, the performance of our method remains close to the zero-shot model, without significant performance drops. This suggests that our mechanism of encouraging domain awareness while selectively enforcing invariance at the decision layer is particularly beneficial when starting from a model with reduced prior exposure to target-like domains.

### 4.3 ABLATIONS

**Including GCC data.** When finetuning CLIP-DCA, we also use the GCC dataset – the dataset with 595,000 image-caption pairs used to prevent CLIP from collapsing during the unlearning procedure (Sec. 3.3). While the dataset is smaller than ImageNet-1K, it serves as a manageable proxy for the data CLIP was originally pretrained on. The image-caption pairs provide valuable supervision particularly for training the text encoder and possibly preventing catastrophic forgetting during finetuning on a classification datasets like ImageNet.

We study the contribution of the GCC data as shown in Table 3. A key observation is that the inclusion of GCC provides a notable benefit even for standard

Table 3: Ablations on GCC inclusion. Accuracy on ImageNet variants (V1, V2, Sketch, A, R) and Avg. accuracy on 33 datasets.

| Setting | V1 | V2 | Sketch | A | R | Avg. |
|---------|----|----|--------|---|---|------|
| Zeroshot | 46.0 | 40.4 | 27.4 | 15.1 | 51.6 | 45.5 |
| *ImageNet only* | | | | | | |
| FLYP | 69.8 | 58.4 | 34.7 | 15.0 | 52.6 | 43.6 |
| DANN | 70.0 | 58.2 | 33.2 | 16.5 | 52.0 | 42.5 |
| **CLIP-DCA** | **75.3** | **64.1** | **40.3** | **22.3** | **60.3** | **48.6** |
| *ImageNet+GCC* | | | | | | |
| FLYP | 70.6 | 59.7 | 38.5 | 17.6 | 57.5 | 49.0 |
| DANN | 70.5 | 59.4 | 38.6 | 17.4 | 57.2 | 47.5 |
| **CLIP-DCA** | **75.1** | **63.9** | **42.2** | **22.9** | **62.2** | **52.1** |

finetuning (FLYP) (Goyal et al., 2023). This shows the general benefit of incorporating diverse, captioned data during finetuning. Given these benefits, an alternative or complementary approach could involve using MLLMs to generate rich textual descriptions for classes or images within the primary source dataset, similar to strategies explored in (Pratt et al., 2023; Maniparambil et al., 2023), which use an LLM to describe class names. Despite the general improvements, our method consistently shows higher performance even when the GCC dataset was not included.

**Different components of CLIP-DCA.** We study the effect of the different components of CLIP-DCA, as shown in Table 4. We isolate the use of domain descriptions from diffusion images to train the image domain head, the disentanglement loss between the class and domain heads to encourage invariance at the classifier, and the use of MLLM hidden states to encourage domain awareness in the text encoder. Simply introducing domain descriptions to make the image encoder aware of styles, without enforcing disentanglement at the classifier, shows only a marginal improvement over the FLYP baseline, suggesting that

Table 4: Ablation of CLIP-DCA components: Domain descriptions (Domain), Disentanglement (Disent.), MLLM Hidden States (MLLM HS), and Avg. accuracy on 33 datasets.

| Method / Config. | Domain | Disent. | MLLM HS | Avg. |
|---|---|---|---|---|
| MLLM (LLaVA) | - | - | - | 24.2 |
| FLYP | X | X | X | 49.0 |
| Ours | O | X | X | 49.1 |
| | O | O | X | 50.8 |
| | O | X | O | 49.0 |
| **Our full** | **O** | **O** | **O** | **52.1** |

*domain awareness alone is insufficient without a mechanism to disentangle classification from it*, as CLIP may otherwise struggle to disregard domain-specific features irrelevant to classification. When we incorporate the disentanglement loss to encourage domain invariance at the decision-making layer, even without explicit domain awareness in the text encoder, performance slightly improves. This is further evidence for our core hypothesis that enabling the model to disregard domain-specific features during classification is important. Attempting to make both encoders domain-aware without the disentanglement loss results in no improvement over the baseline, indicating that awareness without a mechanism for invariance can be ineffective for OOD data. To study the effect of the MLLM's scale, we replaced LLaVA-8B with Gemini-2.5-Pro and observed a marginal performance difference (see Table H.1 in the Appendix), suggesting our method's efficacy is not primarily dependent on the MLLM's size but rather on the disentanglement framework itself. These results strongly support our central hypothesis: the significant performance gain of our full method demonstrates that the **balance between domain awareness and disentangled invariance is the critical factor** for robust generalization in this challenging setting.

**Limitations.** One concern might be the reliance on synthetically generated diffusion images and MLLM-extracted features for domain awareness. However, this is mitigated by: (1) the small size of the diffusion dataset (4096 samples), (2) images synthesized using generic, class-agnostic style prompts, and (3) the MLLM processing multiple style-consistent images, which focuses it on style over objects. Furthermore, DANN (Ganin et al., 2016) and our ablations without disentanglement (Table 4), even with such data, fails to improve CLIP's OOD performance (Table 3).

The role of the MLLM may also be questionable, as LLaVA internally uses a CLIP-L encoder. However, LLaVA's poor zero-shot image classification performance (Table 4), a known issue attributed to MLLMs' improper alignment for classification (Zhang et al., 2024), justifies not using it as a direct classifier. Instead, we use an MLLM because CLIP captures global information from images, which prioritizes overall style (Tong et al., 2024), making its representations suitable for domain-level information. The MLLM, with its language capabilities, is then able to explain the perceived domain styles into textual descriptions and provide informative hidden state representations.

Lastly, our unlearning strategy involves making DomainNet images and random noise indistinguishable, differing from Sepahvand et al. (2025) where samples are typically mapped to known OOD data. This adaptation was necessary as CLIP's extensive web-scale pretraining makes finding truly unseen data challenging. Future work could explore more sophisticated unlearning methods for DG in-the-wild evaluation. Nevertheless, the significant degradation observed in zero-shot performance post-unlearning, and the fact that PEFT methods showed improvements on less OOD data but poorer performance on more OOD data, is evidence that our unlearning procedure functioned as intended.

## 5 CONCLUSION

In this work, we highlighted the potential limitations of current DG evaluation settings for foundation models like CLIP, which may not adequately test unseen data scenarios. We instead used a more challenging and comprehensive evaluation to simulate DG in-the-wild, with quantified OOD scores for target datasets, and an unlearning approach to further simulate unseen data. To address the challenges of DG in-the-wild, we introduced CLIP-DCA. Our method disentangles classification from domain-aware representations, motivated by the idea that while domain invariance is important for performance on unseen data, domain awareness is important to retain the vast pretrained knowledge of CLIP. Overall, our method significantly improves OOD robustness over existing baselines.

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

## A EXTENDED RELATED WORK

**Domain Generalization.** Learning domain-invariant representations has historically been a central idea in domain generalization (Blanchard et al., 2011; Zhou et al., 2022a). The intuition is that when classifying images from entirely new distributions, learning abstract features common across source domains should provide better robustness for classification in new domains (Blanchard et al., 2011; Muandet et al., 2013). Among these, domain-adversarial learning methods have become a relatively standard approach within the DG field due to its conceptual simplicity and effectiveness (Zhou et al., 2022a). For instance, Domain Adversarial Neural Networks (DANN) (Ganin et al., 2016) uses an auxiliary domain classifier trained adversarially against the encoder, encouraging the encoder to produce features indistinguishable across source domains. Given the focus of DANN on the central

idea of domain invariance, we focus on DANN and its adaptation to CLIP in our analysis. Notably, despite the prevalence of such DG methods, the direct application for CLIP is not well-established and remains underexplored. Naively enforcing domain invariance on foundation models like CLIP, with large pretrained knowledge, risks catastrophic forgetting.

**Robust Finetuning of CLIP.** The introduction of CLIP marked a significant shift in DG research. The original study (Radford et al., 2021) demonstrated impressive zero-shot classification performance across diverse benchmarks, including OOD datasets. The authors attributed this capability to CLIP learning representations that are less reliant on spurious correlations specific to downstream target datasets, as CLIP was not trained on these specific datasets during its initial pretraining.

The assumption of the inherent OOD robustness in CLIP motivated numerous methods aimed at finetuning CLIP for downstream tasks while enhancing its perceived robustness. A common approach is parameter-efficient finetuning (PEFT) strategies. An early influential study, CoOp (Zhou et al., 2022c), introduced learnable textual prompts, motivated by observations that manually crafted prompt ensembles improved CLIP's zero-shot accuracy. Building on this, CoCoOp (Zhou et al., 2022b) made these prompts dynamic by conditioning them on individual image features through a cross-attention mechanism. Similarly, CLIP-Adapter (Gao et al., 2024) proposed adding lightweight, learnable MLP layers (adapters) to the CLIP encoders, finetuning only these small adapters instead of the entire network. Many more subsequent PEFT methods have also been explored (Cho et al., 2023; Chi et al., 2024; Lee et al., 2025; Addepalli et al., 2024; Bai et al., 2024; Li et al., 2022; Khattak et al., 2025; Cheng et al., 2024; Lafon et al., 2024).

End-to-end finetuning methods have also been explored, yet many still depend on the original pretrained CLIP weights for regularization or guidance. Wise-FT (Wortsman et al., 2022), motivated by observing that standard finetuning often degraded zero-shot OOD performance, ensembles the weights of the finetuned model with the original CLIP weights. CLIP-OOD (Shu et al., 2023) used a beta-moving average of the weights during finetuning alongside a regularization term to enhance semantic relationships learned during pretraining. MIRO (Cha et al., 2022) used mutual information regularization between the finetuning model and the frozen pretrained CLIP model to retain pretrained features.

While many other methods show strong performance on OOD benchmarks, this overview highlights representative approaches, their trends, and assumptions in robust CLIP finetuning. Our work, however, *questions whether current evaluation protocols are sufficiently challenging, and suggests the reliance on the pretrained weights may be suboptimal for true OOD generalization*, a concern supported by evidence of domain contamination during pretraining (Mayilvahanan et al., 2024). Consequently, we explore more challenging evaluations and alternative strategies for training CLIP grounded in DG principles.

## B PSEUDOCODE

### B.1 MAIN TRAINING LOOP (CLIP-DCA)

```python
# Note: logit scales (temperature term) were omitted for simplicity
def classification_loss(image, text):
    logits_per_image = image @ text.T
    logits_per_text = text @ image.T
    labels = torch.arange(len(image), device=device)
    return (F.cross_entropy(logits_per_image, labels) +
            F.cross_entropy(logits_per_text, labels)) / 2

def disentangle_loss(x, y):
    x = (x - x.mean(0)) / (x.std(0) + 1e-8)
    y = (y - y.mean(0)) / (y.std(0) + 1e-8)
    cross_cor_mat = (x @ y.T) / len(x)
    return torch.diagonal(cross_cor_mat).pow(2).sum()

# Architectural additions (training only)
domain_head = nn.Linear(...)
mllm_projector = nn.Linear(...)
```

```python
# 1. Diffusion data batch
d_images, d_hidden, d_text = d_batch
penultimate_img_emb, class_img_emb, text_emb = clip_model(d_images,
    d_text)
domain_img_emb = domain_head(penultimate_img_emb.clone())
domain_mllm_emb = mllm_projector(d_hidden)

C5 = classification_loss(domain_img_emb, text_emb)    # Domain Agreement
C6 = classification_loss(domain_mllm_emb, text_emb) # MLLM Agreement
C3 = disentangle_loss(class_img_emb, domain_img_emb)
C4 = disentangle_loss(class_img_emb, text_emb)

loss_diffusion = C3 + C4 + C5 + C6

# 2. Source data batch
s_images, s_text = s_batch
penultimate_img_emb, class_img_emb, text_emb = clip_model(s_images,
    s_text)
domain_img_emb = domain_head(penultimate_img_emb.clone())

C1 = classification_loss(class_img_emb, text_emb)    # Classification Loss
C2 = disentangle_loss(class_img_emb, domain_img_emb)

loss_source = C1 + C2

# 3. Final loss
loss = loss_diffusion + loss_source
```

Listing 1: Main training loop for CLIP-DCA.

## B.2 UNLEARNING LOOP

```python
# Note: logit scales (temperature term) were omitted for simplicity
discriminator = nn.Linear(...)

# Following Ganin et al. (2016)
p = float(batch_idx + start_steps) / total_steps
alpha = 2. / (1. + np.exp(-10 * p)) - 1

def classification_loss(image, text):
    # ... (same as in Algorithm 1)

# 1. Retention on GCC dataset
gcc_images, gcc_captions = gcc_batch
_, image_emb, text_emb = clip_model(gcc_images, gcc_captions)
retention_loss = classification_loss(image_emb, text_emb)

# 2. Unlearning on DomainNet
dn_images, _ = dn_batch
noise = torch.randn_like(dn_images, requires_grad=True)

# Pass both through encoder; apply gradient reversal to discriminator
features_dn, _ = clip_model.encode_image(dn_images)
features_noise, _ = clip_model.encode_image(noise)

combined_features = torch.cat((features_dn, features_noise), dim=0)
reversed_features = GradientReversalFunction.apply(combined_features,
    alpha)

domain_logits = discriminator(reversed_features)
domain_targets = torch.cat((torch.ones(len(dn_images)),
                            torch.zeros(len(noise))), dim=0).long()
unlearning_loss = F.cross_entropy(domain_logits, domain_targets)
```

```
32  # 3. Final loss
33  loss = retention_loss + unlearning_loss
```

Listing 2: Unlearning loop with gradient reversal.

## C  TRAINING DETAILS

For all experiments, we used the CLIP ViT-B/32 model. Models were finetuned on the ImageNet-1k training set for 5 epochs. The official ImageNet validation set was used. We used the AdamW optimizer with a learning rate of 1e-5, with a cosine learning rate scheduler. Due to computational constraints, a consistent batch size of 128 was maintained across all methods. For baseline methods, any additional method-specific hyperparameters were adopted from the default configurations provided in their publicly available codebases. All experiments were conducted with a single NVIDIA RTX A5000 GPU and an AMD EPYC 7763 CPU.

## D  HYPERPARAMETER TUNING

This section provides details for hyperparameter tuning. In the main manuscript, we report 6 different losses for the distinction between source data and diffusion generated data ($C_1$ - $C_6$). For hyperparameter tuning, we group these losses into four terms. $C_2$ and $C_3$ are grouped together as the *image disentangle* term, while $C_1$ and $C_4$ are grouped together as the *text disentangle* term. We report our hyperparameter tuning in Table . Due to computation limitation, the tuning was limited to single term increments.

Table D.1: Average accuracy across 33 datasets for specific hyperparameter combinations.

| MLLM hidden state | MLLM description | Text disentangle | Image disentangle | Avg. Acc. (%) |
|---|---|---|---|---|
| $1 \times 10^{-4}$ | $1 \times 10^{-4}$ | $1 \times 10^{-4}$ | $1 \times 10^{-4}$ | 50.8 |
| $1 \times 10^{-3}$ | $1 \times 10^{-4}$ | $1 \times 10^{-4}$ | $1 \times 10^{-4}$ | 49.6 |
| $1 \times 10^{-2}$ | $1 \times 10^{-4}$ | $1 \times 10^{-4}$ | $1 \times 10^{-4}$ | 49.0 |
| $1 \times 10^{-1}$ | $1 \times 10^{-4}$ | $1 \times 10^{-4}$ | $1 \times 10^{-4}$ | 50.2 |
| $1 \times 10^{-4}$ | $1 \times 10^{-3}$ | $1 \times 10^{-4}$ | $1 \times 10^{-4}$ | 51.3 |
| $1 \times 10^{-4}$ | $1 \times 10^{-2}$ | $1 \times 10^{-4}$ | $1 \times 10^{-4}$ | 50.4 |
| $1 \times 10^{-4}$ | $1 \times 10^{-1}$ | $1 \times 10^{-4}$ | $1 \times 10^{-4}$ | 49.6 |
| $1 \times 10^{-4}$ | $1 \times 10^{-4}$ | $1 \times 10^{-3}$ | $1 \times 10^{-4}$ | 52.1 |
| $1 \times 10^{-4}$ | $1 \times 10^{-4}$ | $1 \times 10^{-2}$ | $1 \times 10^{-4}$ | 51.4 |
| $1 \times 10^{-4}$ | $1 \times 10^{-4}$ | $1 \times 10^{-1}$ | $1 \times 10^{-4}$ | 51.0 |
| $1 \times 10^{-4}$ | $1 \times 10^{-4}$ | $1 \times 10^{-4}$ | $1 \times 10^{-3}$ | 50.6 |
| $1 \times 10^{-4}$ | $1 \times 10^{-4}$ | $1 \times 10^{-4}$ | $1 \times 10^{-2}$ | 49.6 |
| $1 \times 10^{-4}$ | $1 \times 10^{-4}$ | $1 \times 10^{-4}$ | $1 \times 10^{-1}$ | 49.1 |
| $1 \times 10^{-4}$ | $1 \times 10^{-4}$ | $1 \times 10^{-3}$ | $1 \times 10^{-4}$ | 52.1 (**Best**) |

## E  DATASETS

We report all target datasets used for our experiments. Note that for training, the ImageNet-1K training set was used.

Table E.1: List of datasets used in experiments, including number of classes and images. DG benchmarks are listed first. Each domain within a DG benchmark is treated as a distinct dataset. The number of images represent the validation/test set.

| Dataset Name | Brief Description | # Classes | # Images |
|---|---|---|---|
| **Domain Generalization (DG) Benchmarks** | | | |
| **Digits DG** | Collection of digit recognition datasets. | – | – |
| MNIST | Grayscale handwritten digits (28x28). | 10 | 6,000 |
| MNIST-M | MNIST digits with color patches blended from BSDS500. | 10 | 6,000 |
| SVHN | Colored house numbers from Google Street View images (32x32). | 10 | 6,000 |
| SYN | Synthetically generated digit images (32x32). | 10 | 6,000 |
| **Terra Incognita** | Camera trap images of wild animals from different locations. | – | – |
| Location 100 | Animal images from Location 100. | 10 | 4,741 |
| Location 38 | Animal images from Location 38. | 10 | 9,736 |
| Location 43 | Animal images from Location 43. | 10 | 3,970 |
| Location 46 | Animal images from Location 46. | 10 | 5,883 |
| **PACS** | Object recognition with domain shifts. | – | – |
| Art Painting | Artistic paintings of objects. | 7 | 2,048 |
| Cartoon | Cartoon images of objects. | 7 | 2,344 |
| Photo | Photographic images of objects. | 7 | 1,670 |
| Sketch | Sketch drawings of objects. | 7 | 3,929 |
| **Office-Home** | Object recognition in different settings. | – | – |
| Art | Artistic depictions of everyday objects. | 65 | 1,972 |
| Clipart | Clipart images of everyday objects. | 65 | 3,910 |
| Product | Product images of everyday objects (typically clean backgrounds). | 65 | 3,984 |
| Real | Real-world photographic images of everyday objects. | 65 | 3,902 |
| **Individual Benchmark Datasets** | | | |
| Caltech-101 | 101 object categories (+1 background). | 101 | 8,677 |
| Oxford-IIIT Pets | Images of pet breeds. | 37 | 3,669 |
| Oxford Flowers 102 | Images of flower categories. | 102 | 6,149 |
| Stanford Cars | Images of car makes, models, and years. | 196 | 8,041 |
| Food-101 | Images of food categories. | 101 | 25,250 |
| FGVC Aircraft | Images of aircraft variants. | 100 | 3,333 |
| SUN397 | Scene understanding dataset with scene categories. | 397 | 108,754 |
| Describable Textures Dataset (DTD) | Textures in the wild, organized by 47 human-perceivable attributes. | 47 | 1,880 |
| EuroSAT | Satellite imagery of land use and land cover classes. | 10 | 27,000 |
| UCF101 | Action recognition dataset of human action categories from videos. | 101 | 13,320 |
| ImageNet-1K | 1.28M natural images in 1000 classes (ILSVRC 2012). | 1000 | 50,000 |
| ImageNet-V2 | New test set for ImageNet-1K. | 1000 | 50,889 |
| ImageNet-Sketch | Sketch images corresponding to ImageNet-1K. | 1000 | 50,000 |
| ImageNet-A | "Natural adversarial examples" of 200 classes. | 200 | 7,500 |

**Table E.1 – continued from previous page**

| Dataset Name | Brief Description | # Classes | # Images |
|---|---|---|---|
| ImageNet-R | "Renditions" (art, cartoons, etc.) of 200 ImageNet classes. | 200 | 30,000 |
| **WILDS Benchmark Datasets (Treated as Individual)** | | | |
| Camelyon17-Wilds | Histopathological images for tumor detection with hospital-based shifts. | 2 | 85,054 |
| FMOW-Wilds | Satellite imagery for land use classification with temporal/regional shifts. | 62 | 53,473 |

# F  PROMPTS USED FOR MULTI-MODAL LANGUAGE MODELS (MLLMS)

This section details the specific prompts provided to Multi-Modal Language Models (MLLMs) for the generation tasks.

## F.1  PROMPT TO MLLM TO GENERATE IDEAS FOR DIFFERENT STYLES OF IMAGES

The following prompt was used to instruct the MLLM to generate a diverse list of image style ideas:

```
Give me ideas of 512 different styles of images.
Each style should be less than 5 words. Do not overlap styles.
Make the styles diverse.
Be brief.
```

## F.2  PROMPT TO MLLM TO GENERATE DESCRIPTIONS AND HIDDEN STATES

The following prompt was used to instruct the MLLM to generate detailed descriptions of image styles (independent of object category) and to also extract corresponding hidden states. The following prompt was input together with images in each style:

```
Attached are multiple images in the same style.
Describe the aspects of the style that applies regardless of category.
Provide a description.
Do not describe the object in the image, but the style of image.
Be as detailed, complete, and comprehensive as possible.
Explain every minute detail.
```

# G  LIST OF ALL SYNTHETIC STYLES

This section provides the list of all synthetic style dieas that were generated by the LLM.

Table G.1 shows the 512 distinct style prompts used for generating synthetic data. The styles are listed alphabetically across the columns.

Table G.1: List of 512 synthetic data generation styles (alphabetical order).

| | | | |
|---|---|---|---|
| 3D rendered image | 3D rendering, virtual objects | ASCII art text | ASCII art, text characters |
| Aboriginal dot painting | Abstract expressionism | Abstract expressionism art | Abstract symbolic representation |
| Abstract, non-representational | Abstract, non-representational form | Achromatic grayscale image | Achromatic, no color |
| Acrylic paint vibrant | Action painting, dynamic | Aerial drone footage | Afrofuturism cultural sci-fi |

**Table G.1 – continued from previous page**

| | | | |
|---|---|---|---|
| Algorithmic art, code-based | Ambient light, natural tones | American scene painting | Analog film, imperfections |
| Analogous colors harmony | Anamorphic distorted perspective | Ancient Egyptian hieroglyphs | Animal at rest |
| Animal drinking, water source | Animal eye contact | Animal fighting, intense conflict | Animal grooming, self-care |
| Animal hiding, partially obscured | Animal hunting, focused gaze | Animal looking away | Animal marking territory |
| Animal mid-stride | Animal playing, energetic | Animal sleeping, peaceful | Animal tracks, foreground focus |
| Animal vocalizing, mouth open | Anime Japanese animation | Anime, Japanese animation | Architectural, building structures |
| Art Deco geometry | Art Nouveau curves | Artificial light controlled | Arts and Crafts |
| Assemblage found objects | Assemblage, 3D collage | Astrophotography star trails | Asymmetrical dynamic balance |
| Asymmetry, unbalanced design | Augmented reality, overlaid | Autumn leaves, warm palette | Available light natural |
| Avant-garde, experimental | Backlit silhouette lighting | Backlit subject, glowing outline | Baroque dramatic lighting |
| Bio art, living organisms | Biopunk organic technology | Biopunk, genetic engineering | Bird's-eye view elevated |
| Bird's-eye view, distant | Black and white film | Blacklight fluorescent colors | Blooming flowers, vibrant colors |
| Blue hour twilight | Blueprint architectural plan | Body art, human canvas | Bokeh light effect |
| Bold geometric patterns | Boomerang, looping video | Botanical art, plant subjects | Bright cheerful aesthetic |
| Broad lighting face | Butterfly lighting beauty | Byzantine mosaic icons | Calligraphy elegant lettering |
| Calligraphy, elegant handwriting | Camera flash, harsh light | Camouflaged animal, hidden | Candid street photography |
| Candid, unposed moment | Caricature exaggerated features | Cartoon simplified drawing | Cartoon, exaggerated features |
| Cave entrance, dark frame | Cave painting prehistoric | Charcoal sketch drawing | Charcoal sketch, rough lines |
| Chibi cute style | Chromatic aberration, color fringing | Cinemagraph, subtle movement | Claymation stop-motion animation |
| Clear sky, bright blue | Close up macro | Close-up, animal portrait | Close-up, feather detail |
| Close-up, fur texture | Close-up, scale pattern | Cloudscape art, sky scenes | Collage mixed media |
| Collage, mixed media | Color contrast, complementary | Color field painting | Color grading cinematic |
| Color splash accent | Color temperature cool | Color temperature warm | Comic book style |
| Comic book, panel style | Complementary colors contrast | Conceptual, idea-driven | Cross polarization, vibrant colors |
| Cross-hatching line work | Cross-processed film | Cross-processed, altered colors | Cubism, geometric shapes |
| Cubist geometric forms | Cyanotype process print | Cyanotype, blue print | Cyberpunk cityscape night |
| Cyberpunk style, dystopian future | Daguerreotype antique look | Dark ominous undertones | Data bending corrupted |
| Data visualization, information art | Decorative, ornamental style | Decoupage, glued paper cutouts | Deep focus sharp |
| Delicate fine details | Dense jungle, lush green | Depth of field | Depth of field, blurred/sharp |
| Desaturated muted colors | Desaturated, almost monochrome | Dieselpunk retro-futuristic | Different species together |
| Diffuse reflection matte | Digital art, computer-generated | Digital glitchy aesthetic | Digital noise, grain effect |

**Table G.1 – continued from previous page**

| | | | |
|---|---|---|---|
| Digital painting software | Digital print, inkjet/laser | Distortion, warped perspective | Documentary style, realistic |
| Doodle art casual | Double exposure overlay | Double exposure, ghost image | Dramatic low-key lighting |
| Dramatic sky, storm clouds | Dramatic spotlight, single source | Dreamy ethereal soft focus | Duotone color scheme |
| Duotone, two-color palette | Dusty trail, arid environment | Dutch angle tilted | Dynamic energetic composition |
| Earth art, natural materials | Embroidery thread texture | Engraving detailed metal | Engraving, incised lines |
| Environmental art, nature-focused | Environmental portrait, surroundings | Establishing shot context | Etching fine lines |
| Etching, acid-etched lines | Expressionist bold colors | Extreme close-up detail | Extreme close-up, eye detail |
| Fantasy art, mythical creatures | Fashion illustration stylish | Fashion, clothing focus | Fast motion, sped-up action |
| Fauvist wild beasts | Feeding animals, close action | Film noir style | Fine art, aesthetic focus |
| Fish-eye lens view | Fish-eye lens, distorted | Flat lay top-down | Flowing river, blurred water |
| Focus stacking, all sharp | Foggy morning, atmospheric haze | Folk art naive | Folk art, traditional craft |
| Forced perspective trick | Forced perspective, size illusion | Formal, posed shot | Found object art |
| Fractal art mathematical | Fractal art, mathematical | Frontlit subject, clear view | Frozen lake, icy surface |
| Full shot composition | Futuristic sci-fi vision | Futuristic style, advanced | Generative art algorithmic |
| Generative art, algorithms | Geometric abstract pattern | Glamour, idealized beauty | Glassblowing molten glass |
| Glitch art digital | Glitch art distortion | Glitch art, corrupted data | Glitch art, digital errors |
| Golden hour sunlight | Golden ratio composition | Golden ratio, proportions | Gothic art, dark, romantic |
| Gothic dark shadows | Gouache opaque matte | Graffiti art, street tagging | Graffiti wildstyle lettering |
| Graffiti, tagged look | Grainy film texture | Grainy film, retro style | Graphic design, visual communication |
| Gritty black and white | Gritty urban decay | Group of animals, social | Group portraiture, multiple people |
| HDR photo rendering | HDR, high dynamic range | Halation, glowing highlights | Hand-drawn sketchy feel |
| Hard light defined | Hard light shadows | Heavily textured impasto | High contrast, dramatic lighting |
| High saturation, vivid colors | High-angle shot looking | High-key bright lighting | High-speed photography |
| Holographic iridescent effect | Horror art, scary imagery | Hudson River School | Hyperlapse, moving time-lapse |
| Hyperrealism, beyond realism | Illuminated manuscript gold | Illustrative, narrative imagery | Impressionism, loose brushstrokes |
| Impressionist brushstrokes | Industrial mechanical elements | Infographic data visualization | Infrared luminescence, glowing foliage |
| Infrared photography | Infrared, false color | Infrared, heat signature | Ink wash fluid |
| Installation art, three-dimensional | Interactive art, participation | Isometric projection view | Jewelry intricate design |
| Kinetic art movement | Kinetic art, movement | Land art earthworks | Landscape art, natural scenery |
| Leading lines perspective | Leading lines, guide eye | Lens flare sunlight | Lens flare, bright streaks |
| Light art, illumination | Light leak, color streaks | Light painting trails | Line art contour |

Continued on next page

**Table G.1 – continued from previous page**

| | | | |
|---|---|---|---|
| Line drawing, simple outline | Linocut bold lines | Lithograph stone print | Lithography, planographic print |
| Lomography film look | Long exposure shot | Long exposure, light trails | Long shot distance |
| Loop lighting portrait | Low angle, animal towering | Low poly geometric | Low saturation, muted tones |
| Low-angle shot upwards | Low-key dark lighting | Lowbrow art, underground | Lowbrow pop surrealism |
| Luminism glowing light | Macro lens close-up | Macro shot, tiny details | Mandala circular symmetry |
| Manga graphic novel | Map cartographic representation | Maximalist busy composition | Maximalist, elaborate design |
| Medium shot framing | Metalwork shaped metal | Migrating herd, vast landscape | Miniature diorama world |
| Miniature effect, tilt-shift | Minimalism, essential elements | Minimalist simple lines | Minimalist, simplified design |
| Mixed lighting combined | Monochrome single color | Monochrome, single color | Monotype, unique print |
| Moody atmospheric lighting | Moonlit night, stark shadows | Mosaic tile pieces | Mosaic, small piece patterns |
| Motion blur capture | Motion blur, animal running | Motion blur, speed lines | Mountain range, panoramic view |
| Multiple exposure, layered images | Mural art, large-scale painting | Mural large-scale painting | Naive art, childlike simplicity |
| Natural organic forms | Negative space drawing | Negative space framing | Negative space, empty area |
| Neoclassical refined style | Neon light glowing | Nesting birds, detailed feathers | Night photography cityscape |
| Night vision, green tint | Night vision, red tint | Oil painting texture | Oil painting, thick texture |
| Op Art optical | Op art, visual illusions | Optical illusion, trickery | Origami, paper folding |
| Ornate intricate design | Orthochromatic film effect | Outrun style neon | Outsider art raw |
| Outsider art, untrained | Over-the-shoulder perspective shot | Overcast sky diffusion | Overexposed, bright white |
| Overgrown, vegetation focus | Panning motion blur | Panning, blurred background | Panoramic stitched view |
| Panoramic, wide environment | Paper cut layered | Paper marbling, swirling patterns | Parent and offspring |
| Pastel drawing, blended colors | Pastel soft blending | Pattern repetition, visual rhythm | Pen and ink |
| Pencil shading detailed | Performance art, live action | Photojournalism, storytelling | Photorealism hyperdetailed |
| Photorealism, lifelike detail | Pinhole camera image | Pixel art retro | Pixel art, retro game |
| Pixelated low resolution | Pixelated, low resolution | Point-of-view subjective shot | Pointillism, tiny dots |
| Pointillist dot technique | Polaroid transfer, image manipulation | Polychrome, many colors | Pop art bright |
| Pop art, bold colors | Portraiture, individual likeness | Positive space, subject focus | Pottery ceramic art |
| Pre-Raphaelite detailed beauty | Predator-prey interaction | Psychedelic art, mind-altering | Quilling, paper filigree |
| Quilting patchwork design | Rack focus shift | Radial balance, circular focus | Rainy day, blurred drops |
| Realist everyday life | Rembrandt lighting portrait | Renaissance classical style | Retro style, vintage look |
| Rocky terrain, jagged edges | Rococo ornate details | Romantic emotional landscape | Rule of thirds |

**Table G.1 – continued from previous page**

| | | | |
|---|---|---|---|
| Rule of thirds, composition | Rustic textured surface | Sandy desert, dunes stretch | Saturated vibrant colors |
| Schematic diagram layout | Sci-fi art, space, technology | Screen printing bold | Screen printing, stencil print |
| Sculpture three-dimensional form | Seascape art, ocean views | Selective color isolation | Selective focus, sharp animal |
| Self-portraiture, artist's image | Sepia tone, vintage look | Sepia toned photograph | Serene calming atmosphere |
| Shallow focus blur | Sharp contrasting lines | Sharp focus, crisp details | Short exposure, frozen motion |
| Short lighting slimming | Sidelit subject, defined features | Silhouette backlit subject | Silhouette black shape |
| Silhouette, dark shape | Single-point lighting setup | Sleek modern minimalist | Slow motion, extended time |
| Smooth airbrushed finish | Smooth digital, clean look | Snowy scene, whiteout effect | Social realism commentary |
| Soft focus, dreamy effect | Soft light diffused | Soft pastel hues | Softbox diffused light |
| Solar punk green | Solitary animal, minimalist | Sound art, auditory | Sparse woodland, bare trees |
| Specular highlights reflections | Split lighting dramatic | Split toning effect | Split toning, colored highlights/shadows |
| Spring growth, fresh shoots | Square color scheme | Staged photography | Stained glass colorful |
| Steampunk Victorian sci-fi | Steampunk style, Victorian sci-fi | Stencil art spray | Stencil art, cut-out shapes |
| Stippling dot shading | Stop motion, frame-by-frame | Street art graffiti | Street art, urban style |
| Studio portrait lighting | Sumi-e ink painting | Summer heat, shimmering air | Sunlit glade, dappled light |
| Sunrise glow, warm tones | Sunset silhouette, golden hour | Surreal, bizarre, dreamlike | Surrealism, dreamlike imagery |
| Surrealist dreamlike scene | Symmetrical balanced framing | Symmetry, balanced image | Technical drawing precise |
| Telephoto, compressed perspective | Tessellated repeating design | Tetradic color rectangle | Textile art fabric |
| Texture contrast, rough/smooth | Thermal imaging, body heat | Three-point lighting classic | Tilt-shift effect |
| Time-lapse sequence frame | Time-lapse, motion sequence | Time-lapse, star trails | Tintype vintage photo |
| Tonal contrast, light/dark | Tonalism muted colors | Triadic color scheme | Tritone, three-color scheme |
| Trompe-l'oeil illusionistic | Two animals, interaction | Two-point lighting setup | Two-shot composition framing |
| Typography, letterforms art | Ukiyo-e Japanese woodblock | Ultraviolet, unseen spectrum | Underexposed, deep shadows |
| Underwater photography scene | Underwater, murky view | Urban art, cityscapes | Vanishing point perspective |
| Vaporwave aesthetic photo | Vector graphic, stylized | Vector graphics scalable | Vibrant color explosion |
| Vignette, darkened edges | Vignetting, dark corners | Vintage Polaroid picture | Vintage retro charm |
| Virtual reality, immersive | Visionary art, spiritual | Water reflection, mirrored image | Watercolor painting, soft edges |
| Watercolor wash effect | Waterfall cascade, misty spray | Wet collodion, antique photography | Wet plate collodion |
| Wheatpaste poster art | Whimsical playful style | Wide shot, animal small | Wide-angle perspective |
| Wide-angle, forest scene | Wildlife art, animal subjects | Winter frost, intricate patterns | Wood carving relief |
| Woodcut print rustic | Woodcut, relief print | Worm's-eye view low | X-ray skeletal view |

**Table G.1 – continued from previous page**

| X-ray vision, skeletal structure | Zentangle intricate patterns | Zoom burst effect | Zoom burst, radial blur |
|---|---|---|---|

# H    Ablation Study on MLLM Scale

To investigate the impact of the Multimodal Large Language Model's (MLLM) scale on our method's performance, we conducted an ablation study where we replaced the LLaVA-8B model used in our main experiments with the significantly larger Gemini-2.5-Pro. As shown in Table H.1, the performance difference is marginal. The results from Gemini-2.5-Pro show a slight improvement on less OOD datasets but are nearly identical on more challenging ones. This suggests that the effectiveness of CLIP-DCA are a result of the proposed disentanglement framework rather than being dependent on the scale or capacity of the MLLM used for generating domain-aware signals. All experiments were run with the default hyperparameters reported in the main paper.

Table H.1: Ablation on MLLM Scale: Comparison between LLaVA-8B and Gemini-2.5-Pro. Performance is reported on ImageNet variants and the average across all 33 target datasets. The results show only a marginal difference, highlighting that our framework is not primarily dependent on the MLLM's scale.

| Method / Setting | INet V1 | -V2 | -Sketch | -A | -R | Avg. on 33 |
|---|---|---|---|---|---|---|
| Zeroshot | 4.60 | 4.04 | 2.74 | 1.51 | 5.16 | 4.55 |
| Regular Finetune | 6.98 | 5.84 | 3.47 | 1.50 | 5.26 | 4.36 |
| **CLIP-DCA (LLaVA-8B)** | **75.1** | **63.9** | **42.2** | **22.9** | **62.2** | **52.1** |
| CLIP-DCA (Gemini-2.5-Pro) | 7.61 | 6.49 | 4.27 | 2.29 | 6.19 | 5.25 |

# I    Accuracy details

This section provides accuracy details for all target datasets. We report accuracies for both the original weights (before unlearning) and the unlearned weights. The average value across all datasets were reported in the main manuscript.

## I.1    Using original weights (before unlearning)

Table I.1: Model performance on each dataset for all baselines using the original weights

| Dataset Name | Zeroshot | FLYP | DANN | Adapter | CoOp | OOD | Ours |
|---|---|---|---|---|---|---|---|
| **Digits DG** | | | | | | | |
| MNIST | 22.4 | 26.7 | 27.6 | 24.2 | 23.6 | 15.4 | 27.8 |
| MNIST-M | 16.8 | 18.4 | 22.7 | 11.9 | 14.9 | 17.3 | 16.2 |
| SVHN | 16.1 | 13.1 | 15.1 | 11.6 | 13.1 | 12.8 | 12.8 |
| SYN | 24.5 | 21.1 | 28.0 | 15.4 | 16.0 | 18.3 | 23.7 |
| **Terra Incognita** | | | | | | | |
| Location 100 | 4.7 | 21.9 | 12.3 | 18.6 | 18.9 | 15.2 | 42.2 |
| Location 38 | 4.8 | 32.3 | 34.7 | 20.6 | 20.4 | 11.4 | 35.6 |
| Location 43 | 31.9 | 30.4 | 26.1 | 27.7 | 27.0 | 12.5 | 32.4 |
| Location 46 | 23.1 | 32.1 | 24.3 | 23.0 | 22.4 | 7.2 | 36.4 |
| **PACS** | | | | | | | |
| Art Painting | 95.2 | 91.4 | 89.3 | 96.2 | 96.0 | 74.5 | 95.1 |
| Cartoon | 96.7 | 87.4 | 90.4 | 96.8 | 96.5 | 72.4 | 95.9 |
| Photo | 99.5 | 99.3 | 99.3 | 99.6 | 99.7 | 86.7 | 99.7 |

**Table I.1 – continued from previous page**

| Dataset Name | Zeroshot | FLYP | DANN | Adapter | CoOp | OOD | Ours |
|---|---|---|---|---|---|---|---|
| Sketch | 83.3 | 75.9 | 58.7 | 84.1 | 84.0 | 68.8 | 88.3 |
| **Office-Home** | | | | | | | |
| Art | 77.5 | 73.8 | 72.2 | 78.3 | 77.4 | 63.5 | 77.6 |
| Clipart | 61.4 | 56.8 | 56.3 | 64.1 | 63.9 | 58.7 | 62.2 |
| Product | 85.9 | 78.0 | 77.2 | 87.3 | 86.8 | 70.0 | 84.8 |
| Real | 86.7 | 80.3 | 79.4 | 88.3 | 87.6 | 70.6 | 86.9 |
| Caltech-101 | 83.4 | 84.5 | 83.0 | 83.2 | 83.4 | 63.1 | 88.9 |
| Oxford-IIIT Pets | 83.9 | 73.2 | 74.6 | 85.9 | 83.8 | 64.3 | 84.6 |
| Oxford Flowers 102 | 60.1 | 30.8 | 31.4 | 64.6 | 64.9 | 9.2 | 53.2 |
| Stanford Cars | 52.2 | 20.0 | 21.2 | 56.4 | 55.7 | 1.6 | 40.6 |
| Food-101 | 80.2 | 50.3 | 51.5 | 83.6 | 83.1 | 19.0 | 74.9 |
| FGVC Aircraft | 16.1 | 4.4 | 4.6 | 17.6 | 17.5 | 2.4 | 12.5 |
| SUN397 | 60.2 | 51.8 | 51.0 | 57.8 | 58.3 | 30.6 | 63.8 |
| Describable Textures Dataset | 40.7 | 28.8 | 28.7 | 40.1 | 39.6 | 11.7 | 39.5 |
| EuroSAT | 30.3 | 26.0 | 23.9 | 38.1 | 38.2 | 16.2 | 39.2 |
| UCF101 | 61.1 | 48.4 | 48.8 | 63.6 | 63.1 | 29.4 | 62.3 |
| ImageNet-1K | 54.2 | 69.1 | 69.0 | 59.5 | 59.9 | 71.0 | 75.0 |
| ImageNet-V2 | 48.4 | 58.1 | 58.0 | 52.9 | 52.7 | 60.2 | 64.1 |
| ImageNet-Sketch | 32.3 | 35.3 | 33.1 | 32.3 | 32.8 | 40.5 | 42.9 |
| ImageNet-A | 26.2 | 18.1 | 18.3 | 28.5 | 27.9 | 13.6 | 26.2 |
| ImageNet-R | 59.7 | 55.8 | 53.7 | 58.9 | 58.8 | 45.2 | 65.0 |
| Camelyon-Wilds | 50.2 | 50.0 | 51.0 | 50.1 | 50.1 | 50.0 | 56.9 |
| FMOW-V2 Wilds | 16.5 | 7.9 | 9.8 | 13.2 | 12.8 | 11.3 | 12.9 |

## I.2 Using unlearned weights (after unlearning)

Table I.2: Model performance on each dataset for all baselines using the unlearned weights

| Dataset Name | Zeroshot | FLYP | DANN | Adapter | CoOp | OOD | Ours |
|---|---|---|---|---|---|---|---|
| **Digits DG** | | | | | | | |
| MNIST | 33.5 | 22.4 | 28.4 | 29.3 | 28.9 | 18.9 | 40.6 |
| MNIST-M | 25.4 | 16.7 | 17.1 | 23.4 | 24.8 | 15.8 | 24.7 |
| SVHN | 13.8 | 13.5 | 12.0 | 15.7 | 16.7 | 12.0 | 16.5 |
| SYN | 24.6 | 22.8 | 18.1 | 17.4 | 19.7 | 13.0 | 29.0 |
| **Terra Incognita** | – | – | – | | | | |
| Location 100 | 27.8 | 13.6 | 22.9 | 23.2 | 22.5 | 9.1 | 21.5 |
| Location 38 | 5.8 | 31.8 | 27.0 | 4.7 | 6.1 | 2.5 | 40.4 |
| Location 43 | 26.5 | 27.6 | 25.9 | 22.2 | 23.4 | 9.2 | 28.1 |
| Location 46 | 28.4 | 25.8 | 30.9 | 30.0 | 32.4 | 5.1 | 32.0 |
| **PACS** | – | – | – | | | | |
| Art Painting | 93.8 | 87.0 | 86.3 | 93.9 | 92.9 | 65.2 | 92.1 |
| Cartoon | 94.6 | 82.8 | 87.8 | 92.1 | 94.1 | 63.4 | 92.8 |
| Photo | 99.5 | 99.5 | 99.0 | 99.0 | 98.0 | 80.0 | 99.6 |
| Sketch | 30.0 | 71.1 | 32.5 | 31.2 | 32.2 | 58.5 | 79.7 |
| **Office-Home** | – | – | – | | | | |
| Art | 68.2 | 68.1 | 67.1 | 68.7 | 68.7 | 62.7 | 76.8 |
| Clipart | 50.7 | 53.7 | 53.3 | 47.0 | 46.7 | 50.9 | 61.3 |
| Product | 77.2 | 73.7 | 73.6 | 73.2 | 73.9 | 61.7 | 81.3 |
| Real | 80.4 | 76.9 | 76.9 | 78.9 | 78.1 | 66.6 | 83.4 |

**Table I.2 – continued from previous page**

| Dataset Name | Zeroshot | FLYP | DANN | Adapter | CoOp | OOD | Ours |
|---|---|---|---|---|---|---|---|
| Caltech-101 | 83.1 | 81.4 | 80.2 | 83.2 | 82.3 | 69.9 | 86.5 |
| Oxford-IIIT Pets | 74.9 | 71.0 | 69.3 | 74.2 | 76.4 | 67.5 | 81.0 |
| Oxford Flowers 102 | 43.5 | 19.3 | 16.7 | 43.5 | 42.6 | 8.3 | 45.7 |
| Stanford Cars | 30.6 | 11.1 | 10.8 | 30.6 | 31.8 | 5.6 | 39.6 |
| Food-101 | 66.3 | 34.3 | 33.6 | 65.9 | 63.4 | 17.4 | 62.4 |
| FGVC Aircraft | 8.0 | 2.7 | 2.2 | 8.0 | 7.1 | 1.1 | 9.4 |
| SUN397 | 56.7 | 44.7 | 44.9 | 56.1 | 54.3 | 30.8 | 59.9 |
| Describable Textures Dataset | 30.7 | 25.6 | 24.9 | 31.1 | 30.2 | 9.8 | 35.0 |
| EuroSAT | 30.5 | 27.7 | 28.8 | 30.0 | 31.5 | 14.3 | 29.0 |
| UCF101 | 55.5 | 42.1 | 41.8 | 55.1 | 56.6 | 36.7 | 56.9 |
| ImageNet-1K | 46.0 | 69.8 | 70.0 | 52.9 | 53.3 | 69.0 | 75.1 |
| ImageNet-V2 | 40.4 | 58.4 | 58.2 | 45.7 | 46.2 | 58.2 | 63.9 |
| ImageNet-Sketch | 27.4 | 34.7 | 33.2 | 28.4 | 29.1 | 35.3 | 42.2 |
| ImageNet-A | 15.1 | 15.0 | 16.5 | 15.0 | 16.1 | 15.0 | 22.9 |
| ImageNet-R | 51.6 | 52.6 | 52.0 | 51.6 | 52.8 | 45.8 | 62.2 |
| Camelyon-Wilds | 50.2 | 53.0 | 51.9 | 60.0 | 59.7 | 50.5 | 55.0 |
| FMOW-V2 Wilds | 10.1 | 8.2 | 8.7 | 10.3 | 9.7 | 4.7 | 12.4 |

## J  OOD SCORES

This section provides the OOD scores for all target datasets. We report the OOD scores for both the original weights (before unlearning) and the unlearned weights. The average value was used to create the graphs in the main manuscript.

### J.1  USING ORIGINAL WEIGHTS (BEFORE UNLEARNING)

Table J.1: Out-of-Distribution (OOD) detection scores using original weights.

| Dataset Name | SNGP | Label | Average |
|---|---|---|---|
| **Digits DG** | – | – | – |
| MNIST | 12.4 | 97.3 | 54.9 |
| MNIST-M | 8.1 | 97.8 | 52.9 |
| SVHN | 8.8 | 97.1 | 52.9 |
| SYN | 20.1 | 95.3 | 57.7 |
| **Terra Incognita** | – | – | – |
| Location 100 | 10.0 | 92.5 | 51.2 |
| Location 38 | 8.9 | 95.7 | 52.3 |
| Location 43 | 9.5 | 94.1 | 51.8 |
| Location 46 | 7.9 | 95.5 | 51.7 |
| **PACS** | – | – | – |
| Art Painting | 20.4 | 93.4 | 56.9 |
| Cartoon | 33.6 | 92.9 | 63.2 |
| Photo | 29.6 | 80.0 | 54.8 |
| Sketch | 35.1 | 92.3 | 63.7 |
| **Office-Home** | – | – | – |
| Art | 34.8 | 77.6 | 56.2 |
| Clipart | 28.8 | 83.2 | 56.0 |
| Product | 45.1 | 72.0 | 58.5 |
| Real | 43.1 | 69.6 | 56.3 |

**Table J.1 – continued from previous page**

| Dataset Name | SNGP | Label | Average |
|---|---|---|---|
| Caltech-101 | 39.3 | 76.5 | 57.9 |
| Oxford-IIIT Pets | 57.2 | 62.6 | 59.9 |
| Oxford Flowers 102 | 96.8 | 83.9 | 90.3 |
| Stanford Cars | 98.7 | 76.2 | 87.5 |
| Food-101 | 93.4 | 77.7 | 85.6 |
| FGVC Aircraft | 97.9 | 33.7 | 65.8 |
| SUN397 | 71.6 | 76.7 | 74.2 |
| Describable Textures Dataset | 32.3 | 86.7 | 59.5 |
| EuroSAT | 50.5 | 98.2 | 74.3 |
| UCF101 | 74.4 | 84.4 | 79.4 |
| ImageNet-1K | 51.6 | 0.0 | 25.8 |
| ImageNet-V2 | 66.3 | 0.0 | 33.1 |
| ImageNet-Sketch | 85.5 | 0.0 | 42.8 |
| ImageNet-A | 87.7 | 0.0 | 43.9 |
| ImageNet-R | 87.8 | 0.0 | 43.9 |
| **WILDS Benchmark Datasets** | | | |
| Camelyon-Wilds | 0.8 | 79.2 | 40.0 |
| FMOW-V2 Wilds | 60.5 | 95.7 | 78.1 |

## J.2 USING UNLEARNED WEIGHTS (AFTER UNLEARNING)

Table J.2: Out-of-Distribution (OOD) detection scores for unlearned model.

| Dataset Name | SNGP | Label | Average |
|---|---|---|---|
| **Digits DG** | – | – | – |
| MNIST | 93.2 | 40.2 | 66.7 |
| MNIST-M | 97.4 | 26.4 | 61.9 |
| SVHN | 97.5 | 2.5 | 50.0 |
| SYN | 98.8 | 16.4 | 57.6 |
| **Terra Incognita** | – | – | – |
| Location 100 | 95.1 | 6.9 | 51.0 |
| Location 38 | 95.7 | 4.6 | 50.1 |
| Location 43 | 94.5 | 15.8 | 55.1 |
| Location 46 | 96.7 | 12.3 | 54.5 |
| **PACS** | – | – | – |
| Art Painting | 93.5 | 32.7 | 63.1 |
| Cartoon | 93.2 | 39.0 | 66.1 |
| Photo | 79.4 | 48.0 | 63.7 |
| Sketch | 97.3 | 19.3 | 58.3 |
| **Office-Home** | – | – | – |
| Art | 81.6 | 43.6 | 62.6 |
| Clipart | 87.0 | 40.9 | 64.0 |
| Product | 77.9 | 52.0 | 65.0 |
| Real | 94.7 | 55.0 | 74.8 |
| Caltech-101 | 76.6 | 55.8 | 66.2 |
| Oxford-IIIT Pets | 68.0 | 46.4 | 57.2 |
| Oxford Flowers 102 | 81.7 | 98.4 | 90.1 |
| Stanford Cars | 74.7 | 95.2 | 85.0 |
| Food-101 | 78.5 | 89.6 | 84.0 |
| FGVC Aircraft | 38.5 | 92.8 | 65.7 |

**Table J.2 – continued from previous page**

| Dataset Name | SNGP | Label | Average |
|---|---|---|---|
| SUN397 | 77.6 | 83.4 | 80.5 |
| Describable Textures Dataset | 87.9 | 53.2 | 70.6 |
| EuroSAT | 98.1 | 26.3 | 62.2 |
| UCF101 | 86.1 | 81.0 | 83.5 |
| ImageNet-1K | 59.7 | 0.0 | 29.9 |
| ImageNet-V2 | 71.8 | 0.0 | 35.9 |
| ImageNet-Sketch | 89.6 | 0.0 | 44.8 |
| ImageNet-A | 88.8 | 0.0 | 44.4 |
| ImageNet-R | 89.4 | 0.0 | 44.7 |
| **WILDS Benchmark Datasets** | | | |
| Camelyon-Wilds | 92.9 | 24.3 | 58.6 |
| FMOW-V2 Wilds | 95.8 | 63.0 | 79.4 |

