# OpenReview forum: "Domain Generalization in-the-Wild: Disentangling Classification from Domain-Aware Representations"
_ICLR.cc/2026/Conference — Submitted to ICLR 2026_

### Official Review · Reviewer_GiyH · 2025-10-31

**Soundness:** 2
**Presentation:** 1
**Contribution:** 2
**Rating:** 2
**Confidence:** 4

**Summary:**

They propose an approach for domain generalization, where they aim to achieve domain-invariant and domain-aware representation learning by using the CLIP model. To achieve it, they proposed to introduce the text2image model and MLLM to generate the images from the specified domains. Empirically, they show the performance improvements in ImageNet variants datasets compared to other domain generalization approaches.

**Strengths:**

1. They show some improvements compared to the baselines they presented.

2. The idea of domain-aware and invariant feature space sounds reasonable.

**Weaknesses:**

1. The effectiveness of their approach is verified only on the ImageNet variants datasets. Probably, the effectiveness is limited to the domains that are easy to describe the styles with words. In addition, the proposed method must know the domain of the target during training, which might contradict the assumption of the DG setting.

2. They need improvements in the presentation of their approach. According to line 196-210, almost all descriptions of their approach are described in the appendix. Since their proposed approach should be the main contribution of this submission, they need to describe it in the main paper. Describing the details in the appendix is okay, yet the authors move almost all parts of the proposed method to the appendix.

3. Overall, the superiority of their approach over other approaches is not very clear. First, the ablation study for their proposed module is not provided. For example, the proposed approach relies a lot on the Stable Diffusion model. However, they do not provide any comparison to other approaches using the text2image model. Then, it is not clear which part of the proposed method contributes to the performance gain.

4. The connection between the observations provided in Sec. 3 and the proposed method in Sec. 2 is not very clear due to the presentation.

**Questions:**

Please respond to the comments on the weakness.

---

> ### Author Response · Authors · 2025-11-21
>
> We thank the reviewer for their constructive comments. We hope to clear up parts of the paper where there may be some misunderstandings and continue a dialogue to continue improving our paper.
>
> &nbsp;
>
> ## *1. Verified only on the ImageNet variants*
> We respectfully clarify that our evaluation is performed on **33 diverse datasets** (detailed in Appendix E), not just ImageNet variants. ImageNet variants represent only a small subset of our "DG in-the-wild" benchmark. Furthermore, our method strictly follows the DG paradigm and does not see target domains during training.
>
> &nbsp;
>
> ## *2. Improvements presentation of approach*
> Thank you for your comment. Are there certain details where the reviewer believes would be useful in the main paper? We will move the pseudocode (Appendix B) to the main paper as space allows. We believe it more effectively conveys our method.
>
> &nbsp;
>
> ## *3. Superiority is not very clear*
> - We respectfully disagree. We respectfully point the reviewer to Table 4 in Section 4.3 (Ablations), which explicitly isolates and evaluates the contribution of each key component: domain descriptions, the disentanglement loss, and MLLM hidden states, concluding that their combination is critical for the observed performance gains.
> - We would also like to respond with two pieces of evidence that suggest the text2image model is not contributing alone to the performance gain.
>   - We see in Table 4, when we only include the domain awareness loss (3rd row), we do not see an improvement over regular finetuning. This suggests synthetic data by itself is not beneficial.
>   - If synthetic data itself drove the improvements, we would expect DANN to also improve since it uses the exact same synthetic images. The fact that DANN performs worse than regular finetuning suggests that it is not synthetic data itself that is driving improvements. We do find an improvement in performance when we increase the number of domains, further suggesting it is the domain diversity that is beneficial:
>
> &nbsp;
>
> ## *4. Connection between Sec. 3 and Sec. 2 is not very clear*
> - Thank you for pointing this out. We acknowledge that if the connection was not clear, our presentation needs improvement. To address this, we will revise the paper to make the link between the problems identified in Section 3 and the solutions in Section 2 more explicit. To clarify the narrative flow: Section 3 establishes the problem of benchmark contamination and inadequate evaluation metrics, while Section 2 introduces CLIP-DCA as the specific solution to these problems. We will revise the transition text between these sections to make this problem-solution link explicit.
>   *   **(Sec 3.1)** Current DG benchmarks may be too simple for CLIP models because different domains in the benchmark are more similar to each other than other datasets (eg. Sketch in PACS is more similar to Photo in PACS compared to Sketch in Office-Home).
>   *   **(Sec 3.2)** Because domains are becoming less discrete in light of web-scale training, we suggest a continuous measure of OODness. We show our metric is accurate comparing the OODness vs performance after finetuning. This metric allows us to show performance on less OOD vs more OOD datasets.
>   *   **(Sec 3.3)** We also propose that to approximate true domain generalization, we use unlearning as a proxy.
>
> - Section 2 introduces our method, CLIP-DCA. Specifically, we believe we are the first to suggest that we can apply domain invariance to foundation models like CLIP. We do this by applying invariance only at the decision making stage, and disentangle it from domain awareness.
>
> &nbsp;
>
> Thank you once again for taking the time to review our paper. We hope we can continue a dialogue to clear up any point that is unclear.

---

### Official Review · Reviewer_bd2c · 2025-10-31

**Soundness:** 2
**Presentation:** 2
**Contribution:** 2
**Rating:** 4
**Confidence:** 4

**Summary:**

This submission studies the challenge of domain generalization (DG) “in-the-wild” for foundation models such as CLIP, noting that existing DG benchmarks are likely contaminated by classes or distributions included in CLIP’s large-scale web pretraining, thus providing an overly optimistic view of OOD robustness. The authors propose two more challenging DG evaluation modes: (1) a broad suite of 33 diverse datasets with a multimodal OOD score and (2) a selective unlearning procedure to “forget” major DG benchmark domains in CLIP. Motivated by observations that naive domain-invariance harms pretrained models, the authors introduce CLIP-DCA: a fine-tuning method that simultaneously enhances domain awareness in the feature encoders and enforces domain-invariance only at the classification layer via disentanglement, leveraging synthetic domains constructed with diffusion models and MLLMs. Their method demonstrates consistent improvements in more challenging OOD settings against standard and robust CLIP baselines.

**Strengths:**

- The paper addresses a critical and underexplored issue in evaluating DG for foundation models, raising urgent concerns about benchmark contamination and the resulting overestimation of generalization.
- The breadth and diversity of the evaluation (33 datasets, supported by a quantitative OOD metric) is a significant step up from most prior works, providing a more rigorous setting for stress-testing OOD robustness.
- The selective unlearning via adversarial training is a clever, tractable workaround for the impracticality of completely removing pretraining contamination, allowing for a controlled analysis of truly unseen domains.

**Weaknesses:**

1.	Many of the claims in the paper are largely empirical and insufficiently supported by theoretical analysis. For example, the statements “enforcing domain invariance could cause catastrophic forgetting” and “naively applying DG methods to foundation models like CLIP can cause catastrophic forgetting” are intuitively plausible but remain anecdotal without a formal theoretical justification or deeper analytical insights.
2.	The proposed dual-head architecture (a classification head and a domain head) is a rather common design pattern. Similar ideas have been explored in prior works such as “Better Pseudo-label: Joint Domain-aware Label and Dual-classier for Semi-supervised Domain Generalization”. As a result, the architectural novelty of CLIP-DCA itself appears limited.
3.	The experimental comparisons rely primarily on older baselines, and the study does not include several more recent DG approaches, even if they use different backbones. Furthermore, the introduction of an additional multimodal large language model (MLLM) component complicates direct comparison, potentially leading to unfair performance advantages relative to standard DG methods.
4.	The writing is occasionally difficult to follow and introduces some unconventional terminology (e.g., “domain awareness”) without adequate explanation or formalization. Several loss functions are described only verbally; providing concise mathematical formulations would make the paper more accessible and rigorous.
5.	The heavy use of CLIP ViT-B/32 as the only model also limits the generality of results.

**Questions:**

The same as Weaknesses.

---

> ### Author Response · Authors · 2025-11-21
>
> We thank the reviewer for their critical, but fair, comments. We hope to address your concerns in our responses.
>
> &nbsp;
>
> ## *1. Claims are largely empirical*
> - We thank the reviewer for this insightful comment. We will explicitly incorporate the following theoretical framework, which builds on findings from Kumar et al. and Zhao et al., into Section 2 of our final revision to ground our empirical findings.
>   - First, standard fine-tuning has been shown to distort pre-trained representations. Kumar et al. (ICLR 2022) theoretically demonstrated that fine-tuning moves model weights away from the robust pre-trained manifold, which Zhang et al. (CVPR 2024) explicitly characterizes as the "catastrophic forgetting" of generic knowledge.
>   - Second, regarding domain invariance, Zhao et al. (ICML 2019) provided a theoretical proof showing that strictly enforcing invariance requires the model to discard discriminative features if they correlate with the domain.
> - Combining these insights, we argue that applying invariance methods to foundation models accelerates the feature distortion identified by Kumar et al. By forcing the model to discard domain-specific features, which are often fine-grained details CLIP learned during pre-training, invariance finetuning could exacerbate the loss of generic knowledge.
>
> &nbsp;
>
> *   "Fine-Tuning can Distort Pretrained Features and Underperform Out-of-Distribution" Kumar et al.
> *   "Overcoming Generic Knowledge Loss with Selective Parameter Update" Wenxuan Zhang et al.
> *   "On Learning Invariant Representations for Domain Adaptation" Han Zhao et al.
>
> &nbsp;
>
> ## *2. Limited novelty of CLIP-DCA*
> - We thank the reviewer for this question. While dual-head architectures are a known design, our main novelty is the hypothesis and implementation to **simultaneously enhance domain awareness in encoders while enforcing domain invariance only at the classifier through disentanglement.**
> - As we mention in lines 129~135, the prevailing method for finetuning CLIP is to use PEFT methods or rely on original pretrained weights for regularization. To the best of our knowledge, we are the first to demonstrate that traditional domain invariance can be beneficial for foundation models like CLIP, provided it is applied only at the classifier level and disentangled from the encoder.
>
> &nbsp;
>
> ## *3. Older baselines and MLLM addition*
> We thank the reviewer for their comment. Our goal was to compare against the most direct and established paradigms for adapting CLIP to new domains, namely PEFT (CoOp, Adapter) and end-to-end regularization (Wise-FT, CLIP-OOD). Many very recent DG methods introduce specialized modules or training schemes designed for different backbones and may not be directly comparable or applicable to a CLIP-centric finetuning setup. We believe our chosen baselines represent the foundational strategies as we mention in lines 676-680. Additionally, we explicitly address the MLLM's role in the limitations section (lines 457-462), justifying its use by demonstrating its poor direct classification performance (Table 4) and its unique ability to extract domain-level information.
>
> &nbsp;
>
> ## *4. Writing is difficult to follow*
> Thank you for these important points. We will add more explanation to the domain awareness term. In short, we use the word as an antonym for domain invariance. We will also add more information for our loss function. As we do provide the pseudocode in Appendix B, we will add this to our main paper if space allows.
>
> &nbsp;
>
> ## *5. Use of CLIP ViT-B/32*
> Thank you once again for your insightful comments. Due to computational constraints in our academic lab, we were limited to the ViT-B/32 model. We acknowledge this as a limitation in the paper. However, we believe our core insight that balancing domain awareness in the encoder with targeted invariance at the classifier is beneficial is a fundamental principle that is not tied to a specific model scale. Our appendix experiment swapping LLaVA for Gemini (Appendix H) provides some initial evidence that our framework is robust to changes in component models.

---

### Official Review · Reviewer_VRcy · 2025-11-01

**Soundness:** 2
**Presentation:** 2
**Contribution:** 2
**Rating:** 2
**Confidence:** 4

**Summary:**

This paper addresses a gap in evaluating domain generalization (DG) for foundation models like CLIP, arguing that current benchmarks may overestimate true out-of-distribution (OOD) robustness due to potential domain contamination in web-scale pretraining data. The authors propose a more challenging evaluation framework termed "DG in-the-wild" consisting of: (1) evaluation on 33 diverse datasets with quantified multimodal OOD scores relative to ImageNet, and (2) unlearning to simulate genuinely unseen domains by making CLIP "forget" specific domains (DomainNet). To address the performance degradation OOD data, the paper introduces CLIP-DCA (Disentangling Classification from enhanced domain Aware representations), which encourages domain awareness in CLIP's encoders while promoting domain-invariant classification through disentanglement at the decision layer. The method uses synthetically generated domain-diverse images from diffusion models and multimodal LLM-generated  descriptions to train a domain head, which is then disentangled from the classification head. Experimental results demonstrate that CLIP-DCA outperforms existing  finetuning methods, particularly on datasets with higher OOD scores and after unlearning.

**Strengths:**

1. **Comprehensive Evaluation:** The experimental study is large and ambitious in scope. Evaluating on 33 diverse datasets, with a quantified multi-modal OOD score for each, provides a broad assessment of model robustness
2. **Originality of unlearning-based evaluation in DG:** The unlearning-based evaluation protocol is innovative, providing a  proxy for assessing genuine OOD generalization without retraining from scratch.
3. **Domain Simulation:** Leveraging diffusion-generated images and MLLM-generated style descriptions to simulate a wide variety of domains is a clever way to sidestep the lack of explicit “domain labels” in most real datasets
4. **Clarity of Presentation:** Overall, the paper is well-organized and generally clear. The inclusion of Figure 2 (illustrating the CLIP-DCA architecture and loss components) and pseudocode snippets in the appendix make the training procedure easier to follow.

**Weaknesses:**

1. **OOD Metric Misalignment with Research Objective:** The paper computes OOD scores based on divergence between the source (ImageNet) and target datasets, yet the central research question concerns the **distance between CLIP’s pretraining data and the target domains**. Although CLIP’s pretraining data are not publicly available, this current OOD metric risks measuring the wrong relationship. The authors could adopt or approximate pretraining-target alignment scores (as used in Teterwak et al. ICLR 2025) to provide a more meaningful quantification of generalization difficulty.
2. **Questionable Justification for the Unlearning Framework:** While innovative, the “unlearning” procedure’s validity as a domain generalization (DG) evaluation strategy is debatable. It is unclear whether adversarially erasing domain information from CLIP truly mimics encountering unseen domains. More direct approaches such as retraining on a smaller surrogate dataset (e.g., ImageNet-1k) with domain-excluded data could provide a cleaner baseline. Without this justification, the realism and interpretability of the proposed evaluation setup remain uncertain.
3. **No Comparison with Alternative Evaluation Strategies:** The proposed unlearning framework is not benchmarked against established DG evaluation methods, such as the *in-pretraining (IP)* vs *out-of-pretraining (OOP)* splits introduced in *“Is Large-Scale Pretraining the Secret to Good Domain Generalization” (Teterwak et al., 2025)*. Without such a comparison, it is difficult to see the benefits of the unlearning-based evaluation.
4. **Scalability and Generalization of the Approach:** Unlearning is a coarse proxy for domain knowledge removal and would need to be repeated for each new target dataset. This limits scalability and makes the framework difficult to reproduce or extend. It remains unclear why this iterative, resource-intensive process is preferable to retraining-based or alignment-based evaluation strategies, especially when the latter can generalize across multiple domains with one training pass.

**Questions:**

1. How would this framework generalize to multiple unseen domains? Would each new evaluation require redoing unlearning, and if so, how do you envision this process being adopted as a general benchmark or practical workflow?
2. What is the computational overhead of the method? Generating synthetic images, querying an MLLM for style descriptions, maintaining multiple heads, and optimizing six loss terms likely introduce their own costs. It would be helpful to know for readers who want to extend your method
3. Wouldn’t testing on these unlearned domains (DomainNet) provide stronger evidence that the model can recover or adapt after unlearning? Without assessing performance on the very domains that were “forgotten,” it is hard to verify whether unlearning improves robustness or simply damages useful representations.

---

> ### Author Response · Authors · 2025-11-21
>
> We thank the reviewer for their detailed and well-thought out review. We hope our rebuttal can clarify our novelties in light of your concerns.
>
> &nbsp;
>
> ## *1. Alternative Evaluation Strategies*
> - We thank the reviewer for this crucial suggestion. We acknowledge Teterwak et al. (2024) in our text and have now implemented their exact evaluation protocol to validate our method **without unlearning**.
> - Here we provide results using the **original pretrained weights.** No unlearning is done. We use the exact splits generated by Teterwak et al, to calculate in-pretraining and out-of-pretraining results. We find that our method does provide a consistent improvement in performance especially on out-of-pretraining data.
>
>   | Dataset | Split | Zero-shot | DANN | FLYP | Adapter | CoOp | CLIP OOD | Ours |
>   | :--- | :--- | :---: | :---: | :---: | :---: | :---: | :---: | :---: |
>   | **PACS** | Full | 93.9 | 92.6 | 93.6 | 94.6 | 93.9 | 93.7 | **95.3** |
>   | | In-pretraining | 95.1 | 92.6 | 94.3 | 95.7 | 94.3 | 94.7 | **96.2** |
>   | | Out-of-pre | 82.6 | 82.3 | 83.4 | 84.6 | 81.0 | 84.3 | **87.1** |
>   | **Terra** | Full | 16.2 | 40.7 | 39.8 | 29.2 | 30.3 | **47.7** | 47.0 |
>   | | In-pretraining | 16.4 | 43.8 | 42.1 | 31.9 | 32.1 | **51.5** | 50.3 |
>   | | Out-of-pre | 12.9 | 17.8 | 20.6 | 8.2 | 7.0 | 16.7 | **21.9** |
>   | **VLCS** | Full | 82.4 | 80.5 | 80.6 | 81.9 | 82.3 | **84.2** | 81.2 |
>   | | In-pretraining | **94.0** | 89.3 | 90.0 | 93.1 | 92.7 | 92.1 | 90.4 |
>   | | Out-of-pre | 74.6 | 75.7 | 76.2 | 74.4 | 72.1 | 80.5 | **82.7** |
>   | **Office Home** | Full | 78.2 | 78.8 | 78.5 | 78.6 | 77.3 | **83.5** | 81.0 |
>   | | In-pretraining | 80.9 | 81.2 | 81.1 | 81.3 | 82.0 | **85.8** | 82.2 |
>   | | Out-of-pre | 45.7 | 52.5 | 48.8 | 47.0 | 77.1 | **58.0** | 57.0 |
>   | **DomainNet** | Full | 55.1 | 56.3 | 56.1 | 57.2 | 56.6 | 59.2 | **60.7** |
>   | | In-pretraining | 66.6 | 65.8 | 67.0 | 68.1 | 67.9 | 69.6 | **70.2** |
>   | | Out-of-pre | 23.2 | 24.5 | 23.9 | 24.2 | 22.2 | 26.3 | **27.7** |
>
> &nbsp;
>
> ## *2. OOD Metric Misalignment*
> - We compare our OOD score and the Pretraining Alignment (Teterwak et al.) score for all of our 33 datasets. We find a significant correlation between our two measures. **Pearson’s r = 0.54 (p = <0.01); Spearman’s r = 0.57 (p = <0.01).**
>
> - In our paper, we hint that the boundary between domains of data is becoming increasingly unclear due to the web-scale pretraining. We thus provide a continuous OOD score that correlates strongly to performance after finetuning (Figure 6 & line 273). This metric also has significant correlation to pretraining alignment. We hope the reviewer can agree that having an accurate continuous measure of OODness is beneficial.
>
>   | Dataset | OOD Score (Ours) | Alignment Score | | Dataset | OOD Score (Ours) | Alignment Score |
>   | :--- | :---: | :---: | :--- | :--- | :---: | :---: |
>   | MNIST | 54.9 | 0.24 | | Camelyon | 40.0 | 0.24 |
>   | MNIST-M | 52.9 | 0.24 | | FMOW | 78.1 | 0.29 |
>   | SVHN | 52.9 | 0.24 | | Caltech-101 | 57.9 | 0.29 |
>   | SYN | 57.7 | 0.25 | | Oxford Pets | 59.9 | 0.31 |
>   | Location 100 | 51.2 | 0.25 | | Oxford-Flowers | 90.3 | 0.33 |
>   | Location 38 | 52.3 | 0.25 | | Stanford Cars | 87.5 | 0.32 |
>   | Location 43 | 51.8 | 0.25 | | Food-101 | 85.6 | 0.31 |
>   | Location 46 | 51.7 | 0.25 | | FGVC Aircraft | 65.8 | 0.29 |
>   | Art Painting | 56.9 | 0.26 | | SUN397 | 74.2 | 0.29 |
>   | Cartoon | 63.2 | 0.27 | | DTD | 59.5 | 0.27 |
>   | Photo | 54.8 | 0.28 | | EuroSAT | 74.3 | 0.28 |
>   | Sketch | 63.7 | 0.28 | | UCF101 | 79.4 | 0.28 |
>   | Art | 56.2 | 0.28 | | ImageNet-1K | 25.8 | 0.29 |
>   | Clipart | 56.0 | 0.27 | | ImageNet-V2 | 33.1 | 0.28 |
>   | Product | 58.5 | 0.29 | | ImageNet-Sketch | 42.8 | 0.28 |
>   | Real | 56.3 | 0.29 | | ImageNet-A | 43.9 | 0.25 |
>   | | | | | ImageNet-R | 43.9 | 0.27 |
>
> &nbsp;

---

> > ### Author Response · Authors · 2025-11-21
> >
> > ## *3. Justification for Unlearning*
> > - Thank you for this insightful point. We agree, and we explicitly state that unlearning is used as a "proxy" (line 285) and an "approximation" (line 020) because retraining a foundation model from scratch is computationally prohibitive (lines 284-286).
> > - Our decision to use the unlearned model for our main analysis in Sec 4.2 was driven by an observation in Sec 4.1. With the original CLIP weights, PEFT methods counter-intuitively performed best on the most OOD data (Fig 8). This contradicts the expectation that performance should decrease with larger OODness even if few parameters are changed.
> > - The unlearning procedure, while not perfect, creates a more controlled experimental setting by reducing this contamination. The resulting performance trends (Fig 9), where all methods degrade more clearly on OOD data, aligning better with expectations of domain generalization.
> > - Most importantly, our results on the Teterwak et al. out-of-pretraining splits (see Table above) confirm that our method achieves superior generalization on real unseen data, validating our performance gains beyond the unlearning proxy. Please also note we already have results for experiments using the original pretrained weights (4.1 Finetuning Original Pretrained CLIP).
> >
> > &nbsp;
> >
> > ## *4. Scalability and Generalization of the Approach*
> > We thank the reviewer for this question. Please note that unlearning was performed **only once** on the DomainNet dataset (which is not part of our 33 evaluation datasets) to create a single, unlearned base model (Sec 3.3). All subsequent fine tuning and evaluation experiments for all methods are performed on this same base model, ensuring a fair and scalable benchmark.
> >
> > &nbsp;
> >
> > ## *5. Computational overhead*
> > Thank you for this question. We provide training details, including hardware and batch sizes, in Appendix C (lines 762). The synthetic data generation is a one-time offline cost, and the resulting dataset is very small (4096 images), making it computationally efficient. The unlearning process is also a one-time process. Finally, the loss terms introduce minimal additional GPU memory footprint as the cost functions are applied to mostly different parts of the embeddings, as seen in our pytorch-style pseudocode (Appendix B).
> >
> > &nbsp;
> >
> > ## *6. Testing on unlearned domains (DomainNet)*
> > Thank you for this great suggestion. We agree that testing on the unlearned domains provides strong evidence of a model's ability to adapt. As shown in the table below, our method effectively recovers performance on the "forgotten" domains, whereas methods that regularize heavily towards the original pretrained weights (e.g., PEFT methods) fail to adapt. In contrast, our method, which actively learns to disentangle domain-awareness from classification, demonstrates a much stronger ability to recover performance.
> >
> >   | DomainNet (Unlearned) | FLYP | DANN | Adapter | CoOp | OOD | Ours |
> >   | :--- | :---: | :---: | :---: | :---: | :---: | :---: |
> >   | Clipart | 53.0 | 56.2 | 55.8 | 53.5 | 53.8 | **57.5** |
> >   | Infograph | 34.0 | 38.5 | 38.2 | 34.5 | 34.7 | **39.8** |
> >   | Painting | 47.0 | 51.5 | 51.1 | 47.5 | 47.8 | **52.9** |
> >   | Real | 73.3 | 76.5 | 76.2 | 73.8 | 73.9 | **77.3** |
> >   | Sketch | 45.5 | 52.8 | 52.3 | 46.0 | 46.2 | **55.1** |
> >   | Quickdraw | 0.3 | 4.1 | 3.8 | 0.5 | 0.6 | **5.3** |

---

> > > ### Comment · Reviewer_VRcy · 2025-11-26
> > >
> > > I thank the authors for thoughtfully engaging with the review, especially the work of evaluating their method using the in/out-pretraining evaluation strategy (Teterwak, et al.). The major issues with the remaining contributions remain and these are critical because this paper claims to ***"better assess the performance of CLIP on DG in-the-wild"*** and more broadly wants to address the issue that ***"current DG evaluation may neither be sufficiently challenging nor adequately test genuinely unseen data scenarios."***
> > >
> > > Like (Teterwak, et al.), this paper wants to specifically **know/quantify how much *supposed OOD* domains differ from what was seen during pretraining**. Therefore this papers contribution could either be an alternative/better way to **(1)** know how similar a target dataset is to the pretraining data to better calibrate accuracy scores and/or **(2)** provide a way to *"remove"* the influence of target-like data in the pretrained weights.
> > >
> > > In its current form, the paper does not convincingly achieve either:
> > >
> > > 1. The proposed pretraining-target OOD metric is effectively a *source (ImageNet)–target metric*, not a true *pretraining–target* measure. This substitution does not address the central question of how much the target domain overlaps with the pretraining data. While access to the pretraining data is understandably hard, this is precisely why Teterwak et al. introduce the Alignment Hypothesis as a justifiable surrogate. By contrast, the current paper’s metric reuses existing similarity measures on ImageNet as a stand-in, and the justification *“we find a significant correlation between our two measures”* is helpful but not sufficient to establish validity. A correlation does not demonstrate that the metric is measuring the intended property (i.e., pretraining-target similarity).
> > >
> > > 2. As the authors themselves acknowledge, the unlearning approach is not a reliable or generalizable way to remove the influence of target-like data from pretrained weights. There is currently no reproducible or principled method to identify which samples should be included in the unlearning set, and this significantly limits the method’s applicability as a standardized DG evaluation strategy.
> > >
> > > These limitations are fundamental because the paper seeks to contribute both a method and an alternative evaluation framework for domain generalization. Without resolving how these components validly capture pretraining-target relationships or reliably simulate unseen domains, the claims of an improved evaluation strategy remain difficult to substantiate.
> > >
> > > For these reasons I will maintain my original score

---

> > > > ### Author Response · Authors · 2025-11-26
> > > >
> > > > We thank the reviewer for the continued engagement. However, we are concerned that the evaluation of our primary contribution, the CLIP-DCA method, is being overshadowed by a critique of the evaluation protocol, even after we successfully validated our method using the reviewer's preferred benchmark.
> > > >
> > > > We wish to clarify three critical points:
> > > >
> > > > &nbsp;
> > > >
> > > > ## *1. The Primary Contribution is the Method (CLIP-DCA), not the Metric.*
> > > > - The reviewer states that the paper "wants to specifically know/quantify how much supposed OOD domains differ."
> > > > - This is a misunderstanding of our core objective. As our title ("Disentangling Classification from Domain-Aware Representations") and introduction state, **our primary goal is to propose a method that balances domain awareness with invariance.**
> > > > - The OOD metric and unlearning was a setup to test our method. It is not the end product.
> > > >
> > > > &nbsp;
> > > >
> > > > ## *2. Validation on the Reviewer’s Preferred Benchmark (Teterwak et al.).*
> > > > - To address concerns about our unlearning/metric setup, we implemented the exact protocol requested by the reviewer (Teterwak et al.). Our method (CLIP-DCA) demonstrates superior performance on these "In-Pretraining vs. Out-of-Pretraining" splits.
> > > > - This proves that the efficacy of CLIP-DCA is not an artifact of our unlearning setup. It works on established, rigorous benchmarks. The validity of our method holds true regardless of the debate surrounding the unlearning proxy. (Which, by the way, we had already showed in 4.1 Finetuning Original Pretrained CLIP)
> > > >
> > > > &nbsp;
> > > >
> > > > ## *3. Contradiction regarding the Unlearning Framework.*
> > > > - We are confused by the shift in the reviewer's stance regarding the unlearning framework.
> > > > Initial Review (Strengths): The reviewer praised the "Originality of unlearning-based evaluation... providing a proxy for assessing genuine OOD generalization."
> > > > - Final Comment: The reviewer now argues that "the unlearning approach is not a reliable or generalizable way... limits the method’s applicability."
> > > > - While we acknowledge the unlearning proxy is not perfect, it was initially recognized as an innovative strength. More importantly, because we have now provided results on the Teterwak benchmark that confirm our method's effectiveness without unlearning, the "reliability" of the unlearning proxy should no longer be a blocking factor for accepting the validity of the method itself.
> > > >
> > > > &nbsp;
> > > >
> > > > ## *Conclusion*
> > > > We have demonstrated that CLIP-DCA improves robustness on both our proposed "in-the-wild" benchmark and the reviewer's requested "out-of-pretraining" benchmark. We respectfully ask that the submission be judged on the effectiveness of the proposed method (CLIP-DCA), which remains robust across different evaluation protocols.

---

### Official Review · Reviewer_3kJ7 · 2025-11-01

**Soundness:** 2
**Presentation:** 2
**Contribution:** 2
**Rating:** 6
**Confidence:** 3

**Summary:**

This paper tackles domain generalization for on top of CLIP, arguing that current evaluation benchmarks may overestimate true out-of-distribution robustness due to web-scale pretraining contamination. The authors propose: 1. a more challenging evaluation framework using 33 diverse datasets with quantified OOD scores; 2. an unlearning-based approach to simulate truly unseen domains; 3. CLIP-DCA, a method that enhances domain awareness in encoders while enforcing domain invariance only at the classification layer through disentanglement. The core hypothesis is that domain awareness is a prerequisite for effective domain-invariant classification in foundation models.

**Strengths:**

1. Important Problem Framing: The paper addresses a critical gap in CLIP evaluation;
2. Comprehensive Evaluation Protocol: The use of 33 diverse datasets with multi-modal OOD scores;
3. Practical Approach: Using diffusion models to generate synthetic domain data and MLLMs to extract domain descriptions is creative and addresses the lack of multi-domain data in single-source datasets like ImageNet;

**Weaknesses:**

1. Architectural Complexity vs. Gains: The method introduces multiple components (domain head, MLLM projector, 6 different loss terms, synthetic data generation pipeline), yet the improvements are often modest.
2. Unlearning as Proxy is Questionable: The unlearning procedure maps DomainNet images to random noise, which is fundamentally different from preventing exposure during pretraining. This doesn't simulate "unseen domains" - it creates adversarially confused representations.

**Questions:**

1. Ablation on Synthetic Data: How does performance change if you use only ImageNet with augmentations (e.g., style transfer, severe augmentations) instead of Stable Diffusion? This would test whether domain diversity or synthetic data itself drives improvements.
2. Domain Head Usage: Why is the domain head discarded at inference? Could it provide calibration signals or be used for test-time adaptation?

---

> ### Author Response · Authors · 2025-11-21
>
> Thank you for your overall positive feedback. We give a point-by-point response to your constructive comments.
>
> &nbsp;
>
> ## *1. Architectural Complexity vs. Gains*
> - Thank you for this thoughtful point. Our core idea of balancing awareness and invariance is algorithmically simple. Our PyTorch-style pseudocode in Appendix B fits the entire training loop in under 50 lines. We will emphasize this algorithmic simplicity and the ease of implementation in Section 3 of our revised paper.
> - Additionally, we respectfully point out that improvements are significant on the more challenging OOD datasets especially after unlearning. The flatter slope of our method's best-fit line in Figure 7 shows superior robustness to severe OOD data, and Table 2 shows a substantial performance gain on difficult datasets like ImageNet-A (+7.9%) and ImageNet-R (+9.6%), where other methods struggle (lines 418-422).
> - Finally, as one of your later points suggest, it is likely possible to further improve performance by using some adaptation enhancements using our domain head. We do not explore this option further to avoid distractions from our main novelty of balancing domain awareness and invariance.
>
> &nbsp;
>
> ## *2. Unlearning as Proxy is Questionable*
> - Thank you for this insightful point. We agree, and we explicitly state that unlearning is used as a "proxy" (line 285) and an "approximation" (line 020) because retraining a foundation model from scratch is computationally prohibitive (lines 284-286).
> - While we use unlearning as a controlled proxy to simulate unseen domains, we also provide new results on the Teterwak et al. out-of-pretraining splits (see Response to VRcy) to validate our method without unlearning.
> - With the original CLIP weights (4.1 Finetuning Original Pretrained CLIP), PEFT methods counter-intuitively performed best on the most OOD data (Fig 8). This contradicts the expectation that performance should decrease with larger OODness even if few parameters are changed. The unlearning procedure, while not perfect, creates a more controlled experimental setting by reducing the possible domain contamination. The resulting performance (Fig 9) shows all methods degrade more clearly on OOD data, aligning better with expectations of domain generalization.
>
> &nbsp;
>
> ## *3. Ablation on Synthetic Data*
> - Thank you for this unique thought-provoking view. We would like to respond with two pieces of evidence.
>   - We see in Table 4, when we only include the domain awareness loss (3rd row), we do not see an improvement over regular finetuning. This suggests synthetic data by itself is not beneficial.
>   - If synthetic data itself drove the improvements, we would expect DANN to also improve since it uses the exact same synthetic images. The fact that DANN performs worse than regular finetuning suggests that it is not synthetic data itself that is driving improvements.
> - We do find an improvement in performance when we increase the number of domains, further suggesting it is the domain diversity that is beneficial:
>
>      | Number of domains | Average performance on 33 datasets |
>      | :--- | :---: |
>      | 32 domains (256 images) | 48.5 |
>      | 64 domains (512 images) | 49.0 |
>      | 128 domains (1024 images) | 49.9 |
>      | 256 domains (2048 images) | 51.8 |
>      | **512 (All domains) (4096 images)** | **52.1** |
>
>
> &nbsp;
>
> ## *4. Domain Head Usage*
> We sincerely thank the reviewer for their questions. That's an excellent suggestion for future work. The potential for using the domain head for calibration or test-time adaptation is a promising direction. In our current framework, we treated it as an auxiliary component to achieve our primary goal of a disentangled, domain-invariant classifier, which is why it's discarded at inference. Namely, we do not want to draw attention away from our main novelty: **balancing domain invariance and awareness**.

---

### Author Response · Authors · 2025-11-21
**Common Response**

We sincerely thank all reviewers for their constructive feedback. We are encouraged that reviewers found our work addresses a "critical gap in CLIP evaluation" (R 3kJ7) and uses a "clever, tractable workaround" (R bd2c) for domain simulation.

We are glad that reviewers recognized several strengths in our work:
*   **Evaluation Framework:** Our "Comprehensive Evaluation Protocol" (R 3kJ7) was noted for its "large and ambitious scope" (R VRcy) and "breadth and diversity" (R bd2c) across 33 datasets.
*   **Unlearning Approach:** The "Originality of unlearning-based evaluation in DG" (R VRcy) was highlighted as a "clever, tractable workaround" (R bd2c) for simulating truly unseen domains.
*   **Methodology:** Our "Practical Approach" (R 3kJ7) of using diffusion models for "Domain Simulation" (R VRcy) was seen as a creative solution. The core "idea of domain-aware and invariant feature space" was also found to be reasonable (R GiyH).

&nbsp;

We would also like to address some common concerns, particularly regarding the novelty of our method.

&nbsp;

## *Novelty of CLIP-DCA*
Our contribution is the idea enabled by the dual-head architecture, not the architecture itself. We believe we are the first to successfully **apply domain invariance to the original CLIP parameters**. Prevailing methods for finetuning CLIP either regularize towards the original weights or use parameter-efficient methods. Our key insight and novelty is to show that invariance can be beneficial if it is applied to the classifier, while domain awareness (the opposite of invariance) is explicitly preserved in the encoder.

&nbsp;

## *Justification for Unlearning*
We acknowledge the concerns that unlearning is an imperfect proxy/approximation for a truly unseen domain (R 3kJ7, R VRcy). We agree and state this in the paper (lines 20, 285). Its purpose is to create a more controlled experimental setting. However, **please also note that we do test with the original weights (4.1 Finetuning Original Pretrained CLIP).**

As an alternative evaluation suggested by Reviewer VRcy, we test on the out-of-pretraining split on the DomainBed dataset. We find that our method shows higher performance on the out-of-pretraining split.

| Metric | Zero-shot | DANN | FLYP | Adapter | CoOp | CLIP OOD | Ours |
| :--- | :---: | :---: | :---: | :---: | :---: | :---: | :---: |
| **Out-of-pretraining (avg)** | 47.8 | 50.6 | 50.6 | 47.7 | 51.9 | 53.2 | **55.3** |

&nbsp;

## *Presentation and Clarity*
We appreciate the feedback on improving the clarity of our paper (R bd2c, R GiyH). In the revised version, we will move some details, such as our pseudocode, into the main paper. We will better formalize terms like "domain awareness" to make our contributions more rigorous and accessible.

---

### Meta-Review · Area_Chair_fWgY · 2026-01-05

**Summary:**

The submission received mixed scores (6, 2, 4, 2) with significant pushback on the evaluation protocol. Reviewer VRcy (score 2) strongly argued that using "unlearning" to simulate unseen domains is not a reliable proxy. Reviewer GiyH (again score 2) criticized the presentation, noting key methodological details were buried.

**Reviewer Concerns:**

Although the authors added the requested "out-of-pretraining" benchmark, Reviewer VRcy maintained that the proposed OOD metric is effectively a standard one and fails to address pretraining contamination. Additionally, the presentation issues identified by GiyH regarding the clarity of the novel method remain largely unresolved.

**Reviewer Scores:**

Reviewer VRcy explicitly maintained their reject rating post-rebuttal. Reviewers GiyH and bd2c did not have a chance to engage post-rebuttal but would likely maintain their low/marginal scores due to the outstanding issues with presentation and novelty. The best-case scenario would be for the marginally positive reviewer (3kJ7) to uphold their rating, which will still render a holistically negative evaluation of the paper.

---

### Decision · Program_Chairs · 2026-01-26

Reject